# Diurnal oscillations in gut bacterial load and composition eclipse seasonal and lifetime dynamics in wild meerkats

Alice Risely ⬤ [1✉], Kerstin Wilhelm ⬤ [1], Tim Clutton-Brock ⬤ [2,3,4], Marta B. Manser ⬤ [3,4,5] & Simone Sommer ⬤ [1]

Circadian rhythms in gut microbiota composition are crucial for metabolic function, yet the extent to which they govern microbial dynamics compared to seasonal and lifetime processes remains unknown. Here, we investigate gut bacterial dynamics in wild meerkats (*Suricata suricatta*) over a 20-year period to compare diurnal, seasonal, and lifetime processes in concert, applying ratios of absolute abundance. We found that diurnal oscillations in bacterial load and composition eclipsed seasonal and lifetime dynamics. Diurnal oscillations were characterised by a peak in *Clostridium* abundance at dawn, were associated with temperature-constrained foraging schedules, and did not decay with age. Some genera exhibited seasonal fluctuations, whilst others developed with age, although we found little support for microbial senescence in very old meerkats. Strong microbial circadian rhythms in this species may reflect the extreme daily temperature fluctuations typical of arid-zone climates. Our findings demonstrate that accounting for circadian rhythms is essential for future gut microbiome research.

[1] Institute for Evolutionary Ecology and Conservation Genomics, Ulm, Germany. [2] Large Animal Research Group, Department of Zoology, University of Cambridge, Cambridge, UK. [3] University of Pretoria, Mammal Research Institute, Pretoria, South Africa. [4] Kalahari Research Trust, Kuruman River Reserve, Northern Cape, South Africa. [5] Department of Evolutionary Biology and Environmental Studies, University of Zurich, Zurich, Switzerland.
✉email: alice.risely@uni-ulm.de

Gut microbial communities are highly dynamic and rapidly respond to factors such as host diet and immunity[1,2]. These responses can generate predictable community dynamics that act at varying time scales[3,4], including circadian rhythms triggered by light-dark cycles and food intake[5–7], seasonal shifts in response to food availability and climate[8–10], and predictable patterns in microbiome maturation and senescence across host life[11,12]. However, these processes have only been studied independently rather than collectively. Hence, the relative importance of interacting temporal scales to governing gut microbiome dynamics remains poorly understood. This gap limits our ability to identify associations between the gut microbiome and host physiology and fitness. For example, accounting for age-related shifts in the gut microbiome improves the detection of disease-microbiome interactions[13].

Circadian rhythms in gut microbiomes remain particularly understudied, despite their role in maintaining host physiological homoeostasis[7,14–16]. In laboratory mice, microbial circadian rhythms are characterised by a spike in bacterial load at dusk when they become active and begin feeding[6,17], with members of Clostridiales becoming particularly inflated[5,6,17–19]. These diurnal oscillations trigger a cascade of physiological changes to the host via alterations to the metabolome and cell transcription patterns[6], and modulate immune function and pathogen susceptibility across the day[16]. Whilst control of microbial circadian rhythms lies largely with host circadian clock genes[17], both irregular diets and prolonged dark exposure disrupt diurnal rhythms[18,19]. Notably, these studies have almost exclusively been conducted in laboratory animals. The exception is a study that identifies weak diurnal signatures in relative abundances in a large cohort of humans, where time of day explained 0.1% of microbial composition[7]. This raises the question whether circadian rhythms in gut microbiomes of wildlife reflect those found in model systems, in particular when accounting for bacterial load.

In addition to diurnal oscillations, gut microbiome composition changes with season and host age. Whilst seasonal fluctuations are prevalent across studied species and their function relatively well documented[8–10], the development and senescence of the gut microbiome over host life remain elusive. In humans, microbiome alpha diversity increases over infancy[11], whereas it decreases in chimpanzees[20], although the gut microbiome of infants tends to have higher inter-individual variation in both species[20]. In old age, the gut microbiome of humans and model animals becomes depleted in core taxa[12,21,22]. Another potential characteristic of the aged microbiome may be a decline in microbial circadian rhythms and subsequent dysbiosis[23]. Since physiological circadian rhythms and their associated behaviours decay with age[24–26], it is conceivable that altered hormonal cycles are reflected in dampened gut microbiome diurnal rhythms, yet this has so far not been investigated. Crucially, the identification of predictable dynamics over host life requires long-term and longitudinal datasets of known individuals, yet in addition depends on a robust understanding of the short- and medium-term dynamics that shape gut microbial communities over hours, days and months. To date, no studies of gut microbiome dynamics have accounted for interacting temporal scales ranging from hours to years, nor incorporated fine-grain behavioural and environmental data to understand the underlying mechanisms.

To investigate how interacting diurnal, seasonal, and lifetime dynamics together shape estimated bacterial load and diversity in a wild host system, and to identify their biological and environmental mechanisms, we analysed 1109 faecal samples collected from 235 wild meerkats (Suricata suricatta) between 1997 and 2019 (mean no. samples per meerkat = 5, min. = 1, max. = 14; Fig. 1a). Meerkats are small insectivorous mongooses that inhabit arid regions of southern Africa, and form social groups of two to fifty individuals led by a dominant pair. The meerkat population investigated here is located in South Africa and has been monitored since 1995 by the Kalahari Meerkat Project, which collects detailed data on body condition, behaviour, and life histories of individually marked individuals[27]. Meerkats from the study population experience a number of biological processes that act at daily, seasonal, and lifetime scales that may be expected to interact with the gut microbiome. For example, daily foraging schedules are temperature constrained, with meerkats foraging most intensely in the early morning and again from mid-afternoon until sunset when temperatures are cool (Fig. 1b). In summer, when day temperatures often reach 40 °C, they begin foraging at sunrise and cease completely during the middle of the day when they rest[28]. The Kalahari region is also highly seasonal, with the climate marked by high temperatures and sporadic rainfall during the wet summer (October to April), and dry winters (May to September; Fig. 1c) being cool with almost no rainfall. Meerkat diet diversity increases in the wet season, yet arthropods make up the majority of the diet throughout the year[28]. Lastly, the timing of relevant life-history stages such as weaning and senescence are well characterised in this population. Pups leave their natal burrow to forage at approximately one-month old and are weaned at approximately 9 weeks. Meerkats reach sexual maturity at nine months, and biological and reproductive functions begin senescing between 5 and 6 years of age[29] (Fig. 1d).

Faecal pellets were collected from individually marked individuals at the point of defecation, and sampling distribution across the three temporal scales is shown in Fig. 1e. Samples were collected mostly between 6 am and 1 pm, and again between 3 pm and 8 pm when meerkats are most active, and were measured with reference to hours after sunrise. Samples were stored next to an icepack and then frozen on return to the field station after either the morning or afternoon field session. For long-term storage, samples prior to 2008 were mostly frozen at −80 °C (n = 461), or, after 2008, freeze-dried and kept at room temperature (n = 648; Supplementary Fig. 1a). Microbiome phylogenetic profiling was performed using 16S ribosomal RNA (rRNA) gene amplicon sequencing of the V4 region to generate Amplicon Sequence Variants (ASVs)[30]. We estimated total 16S copy number, often used as a proxy of bacterial load (i.e. the number of bacterial cells) by scaling reads to an internal standard that was added to each weighed sample prior to DNA extraction.

Here, we use this large longitudinal dataset of a well-studied free-ranging wildlife species to (1) compare the strength of diurnal, seasonal, and lifetime dynamics in bacterial load, alpha diversity, and beta diversity; and (2) to identify which genera exhibit predictable dynamics at each scale. Further (3), we test for any environmental and behavioural mechanisms that may underpin the observed dynamics, and (4) test whether microbial circadian rhythms decay with age. We show that diurnal oscillations are stronger and more prevalent across microbial community and genus-level phenotypes than seasonal or lifetime processes. These oscillations are linked to temperature-constrained foraging schedules, with most common genera spiking at dawn when meerkats begin to feed, and declining at noon when temperatures often become too hot for foraging. Lastly, we show that these diurnal oscillations do not appear to decay in old age.

## Results

**Effects of storage and technical variation.** We first validated our methods by assessing the effect of storage and technical variation on microbiome composition. To quantify the effect of the two storage methods on bacterial composition in fresh samples, we performed a separate pilot study with nine faecal samples sourced

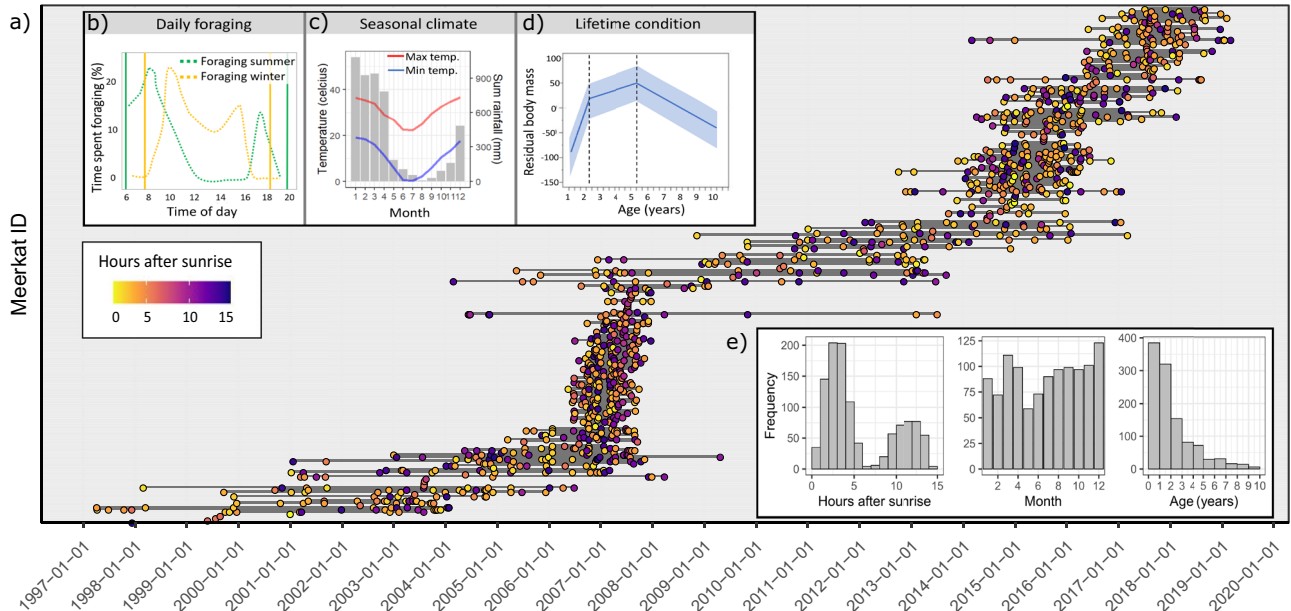

**Fig. 1 Study system and sampling distribution. a** Timeline of samples analysed in this study (1997–2019; $n = 1109$), with lines connecting samples collected from the same meerkat individual (*y*-axis), and coloured by hours after sunrise. Yellow represents samples collected close to sunrise, purple represents samples collected closer to sunset. Periods of intensive sampling (~2007 and 2015) enable us to account for environmental and social effects at certain periods. **b** Proportion of time meerkats spend foraging during wet summer (green dashed line) and dry winter (yellow dashed line). Figure modified from Doolan and MacDonald[28] with permission. Solid lines represent sunrise and sunset in summer (green) and winter (yellow). **c** Seasonal climate across the year measured at the Kalahari Research Station, South Africa, averaged from data between 2009 and 2019. Bars represent total rainfall per month, and red and blue lines represent mean maximum and minimum temperatures, respectively. **d** Trend in residual body mass across life showing senescence at approximately 5.5 years and 95% credible intervals, modified from Thorley et al.[29]. **e** Sampling distribution for diurnal, seasonal, and lifetime scales. Source data are provided in the source data file.

from nine captive meerkats at Zurich University. Samples were immediately frozen after collection, and then either freeze-dried or kept frozen at −80 °C for seven days. Microbiome composition clustered strongly by sample identity in their beta diversity (Supplementary Fig. 1b), and storage did not significantly affect composition (Weighted Unifrac: $F = 0.7$, $p = 0.52$; Unweighted Unifrac: $F = 1.0$, $p = 0.37$). Across samples analysed in this study, storage had significant yet small effects on estimated bacterial load, with frozen samples overall having slightly lower estimated abundance ($t = 7.2$, $p < 0.001$, $R^2 = 0.04$; Supplementary Fig. 1c). Observed ASV richness did not differ between storage types ($t = 0.7$, $p = 0.48$; Supplementary Fig. 1d). Storage had weak but significant effects across four measures of beta diversity ($p < 0.001$, $R^2 = 0.01–0.02$; Supplementary Fig. 1e). Because storage had small effects on the measured composition, we account for storage in all models, and we only consider associations robust if they exhibit significant trends across both frozen and freeze-dried samples.

We tested how micro-variation in weighing and other sources of technical variation affected estimated load and diversity measures by including 16 extraction replicates, which were treated separately at every stage of processing. Technical variation accounted for 10% of variation in estimated bacterial load (Supplementary Fig. 2a), and 1–2% of variation across measures of alpha diversity (Supplementary Fig. 2b) and the first axis of variation of four measures of beta diversity (Supplementary Fig. 2c–f). Whilst technical variation was non-negligible, sample ID accounted for 90–98% of variation across measures.

**Time of day is the strongest predictor for bacterial load and diversity.** We first aimed to investigate how estimated bacterial load, alpha diversity, and beta diversity change over the diurnal,

seasonal, and lifetime scales. We modelled bacterial load and alpha diversity across the three time scales by fitting generalised additive mixed models (GAMMs) to log-transformed bacterial load and (untransformed) ASV richness, applying non-linear smoothing functions to time of day, month, and meerkat age, whilst controlling for sampling depth, sequencing run, storage method, and time in field conditions prior to being frozen as fixed effects, and including individual ID and social group as random effects. Definitions of all terms included in models are outlined in Supplementary Table 1.

Mean bacterial load underwent the largest shifts across the day, in comparison to seasonal and lifetime scales, which were both much weaker (Hours after sunrise: $F = 54.4$, $p < 0.0001$; Month: $F = 1.1$, $p = 0.007$; Age: $F = 9.1$, $p = 0.003$; model $R^2 = 0.47$; Supplementary Table 2). Bacterial load tended to be highest early in the morning and lowest approximately 10 h after sunrise (Fig. 2a), although it should be noted there is considerably uncertainly regarding estimates for the middle of the day when sampling is sparse. Bacterial load fluctuated only weakly with season (Fig. 2b) and age (Fig. 2c). Whilst seasonal and lifetime shifts in bacterial load were weak but significant across the full dataset, they were not replicable across both frozen and freeze-dried samples (Fig. S3a). Sequentially excluding terms to assess the drop in model explanatory power ($R^2$) indicated that diurnal, seasonal, and lifetime dynamics accounted for 12%, 1%, and 1% variation in bacterial load, respectively. Individual ID and social group together accounted for 2% of variation, whilst methodological variables accounted for 31%.

ASV richness also demonstrated the strongest fluctuations across the day (Hours after sunrise: $F = 20.8$, $p < 0.0001$; Month: $F = 3.4$, $p < 0.0001$; Age: $F = 3.8$, $p = 0.072$; model $R^2 = 0.3$; Supplementary Table 3). In contrast to bacterial load, observed richness was lowest in the early morning and evening, and highest

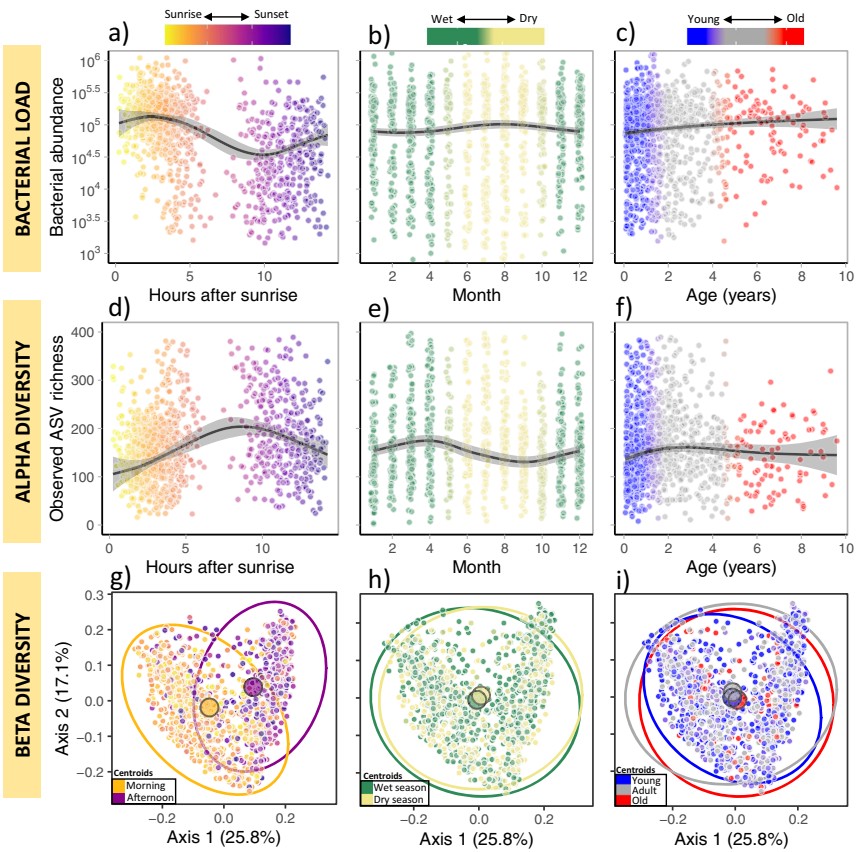

**Fig. 2 Temporal trends in gut bacterial load (top), alpha diversity (middle), and beta diversity (bottom) across the day (left), year (centre), and life (right). a–f** Smoothed estimates and partial residuals from two GAMMs predicting (**a–c**) bacterial load and (**d–f**) observed ASV richness across the day, year, and life. Shaded area represents 95% CIs. **g–i** Beta diversity MDS ordination coloured by (**g**) hours after sunrise (orange = morning; afternoon = purple), (**h**) season (green = wet season; yellow = dry season), and (**i**) age (blue = young; grey = adult; red = old). For clarity, samples were grouped into discrete categories to generate 95% CI ellipses and group centroids (large circles), which represent samples collected in the morning field session (< 7 h after sunrise) and afternoon (> 7 h after sunrise), wet (October–April) and dry (May–September) season, and age categories (< 1 years, 1–5 years, and >5 years). Source data are provided in the source data file.

in the middle of the day (Fig. 2d). Across the year, mean ASV richness increased over the wet summer, and declined over the dry winter (Fig. 2e). ASV richness did not exhibit significant changes across life (Fig. 2f). These results were consistent across frozen and freeze-dried samples (Supplementary Fig. 3b). Overall, our model accounted for 30% of variation in ASV richness. However, diurnal, seasonal, and lifetime dynamics accounted for only 6%, 3% and 1%, respectively; meerkat ID and social group accounted for 4%, whilst methodological variables accounted for 16%. We tested whether these trends were also evident when applying the abundance-weighted Shannon diversity, and found weaker and inconsistent trends across all temporal scales (Supplementary Fig. 3c). Thus, changes to observed ASV richness across the day were driven by rare taxa.

We explored shifts in beta diversity across the three temporal scales by ordinating taxa composition using Multi-Dimensional Scaling (MDS) analysis of Weighted Unifrac distances, which accounts for both abundance and phylogeny. Community composition along the first two ordination axes clustered most strongly by the time of day (Fig. 2g), and only very weakly by season (Fig. 2h) or age (Fig. 2i). A PERMANOVA test on the weighted Unifrac distance matrix indicated that time of day had the strongest effect size yet explained relatively little variation (Hours after sunrise: $F = 49.7$, $R^2 = 0.038$, $p < 0.001$; Month: $F = 3.1$, $R^2 = 0.002$, $p = 0.003$; Age: $5.28$, $R^2 = 0.004$, $p < 0.001$; Supplementary Table 4a). Meerkat ID explained a large proportion of variation yet had a small effect size (i.e. centroids

are close together; $F = 1.1$, $R^2 = 0.19$, $p = 0.032$), whilst social group was not significant ($F = 0.9$, $R^2 = 0.03$, $p = 0.88$). Similar results were generated when using Unweighted Unifrac (Supplementary Table 4b) and across storage types (Supplementary Fig. 3d, e).

**Genus-level dynamics.** We next aimed to identify which genera were influencing shifts in beta diversity and to model the dynamics of genus-level abundances across temporal scales. Changes to beta diversity across the day reflected a decrease in the relative abundance of the genus *Clostridium* between morning and afternoon (Fig. 3a). Axes 1 and 2 of the Weighted Unifrac ordination shown in Fig. 2g–i largely represented the continuum of *Clostridium* and *Bacteroides* abundances, respectively, and sample placement along these two axes was strongly influenced by time of day ($R^2 = 0.27$, $p < 0.001$), but only very weakly by season ($R^2 = 0.004$, $p = 0.02$) and not by age ($R^2 = 0.001$, $p = 0.48$). Samples taken in the morning were more likely to be dominated by *Clostridium* than those taken in the afternoon, which tended to be dominated by a more diverse suite of *Raoultibacter*, *Cellulomonas*, Bacillicaceae, *Enterococcus* and *Lactococcus* (Fig. 3b). In contrast, axes 3 and 4 largely represented the continuum of *Paeniclostridium* and *Blautia* abundances, respectively, and sample placement along these axes was more associated with age ($R^2 = 0.043$, $p < 0.001$) and weak seasonal effects ($R^2 = 0.007$, $p = 0.003$), rather than time of day

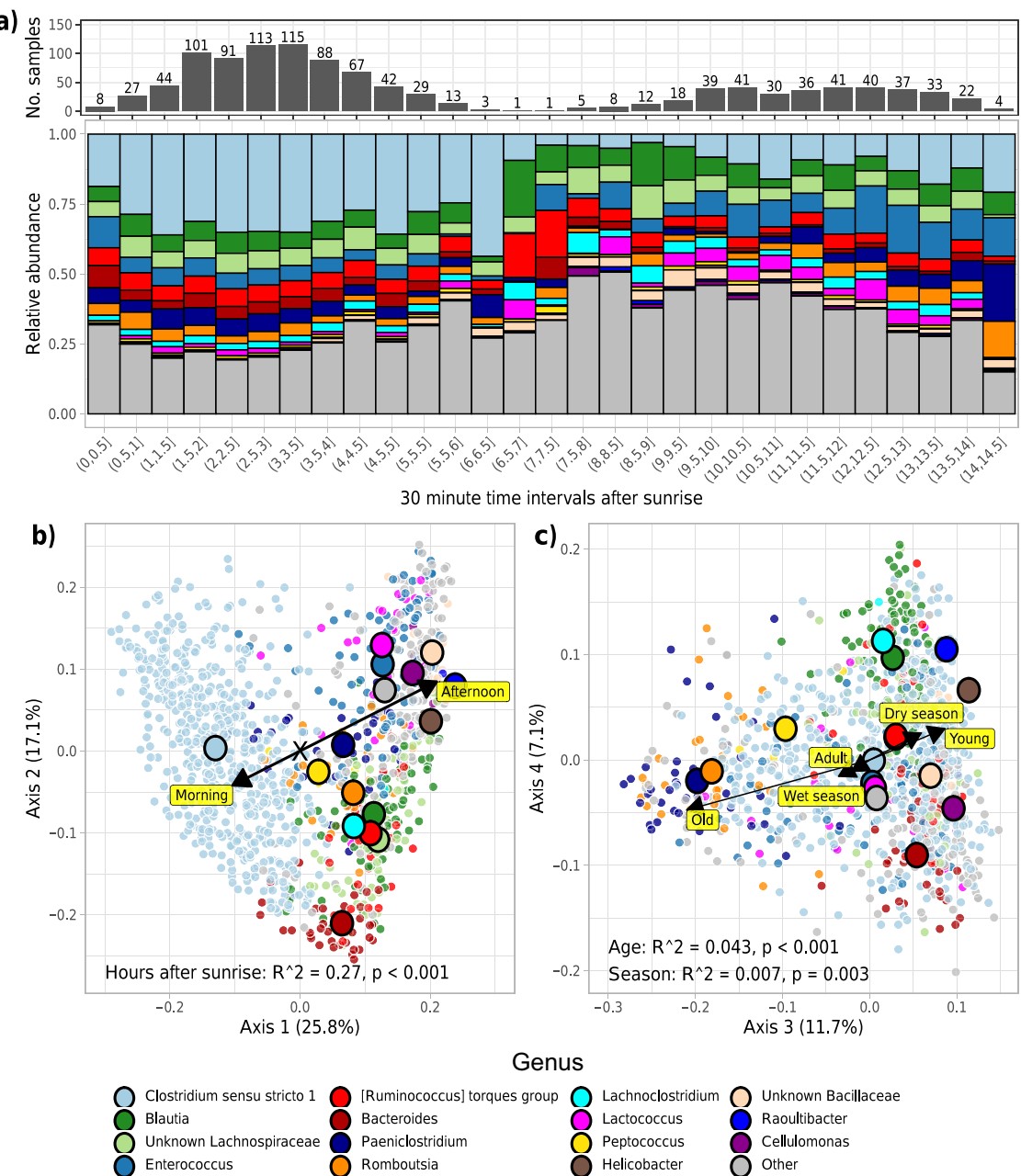

**Fig. 3 Genera driving temporal trends in beta diversity. a** Summary of taxonomic shifts in relative abundance at the genus-level per 30-min interval from sunrise. The number of samples that were summarised per 30-min slot are indicated by the histogram. **b**, **c** Weighted Unifrac ordination plots of (**b**) axes one and two and (**c**) three and four, coloured and grouped by the most abundant genus in each sample. Large circles represent group centroids for samples sharing the same most abundant genus. Arrows indicate the direction and influence of significant temporal variables when categorised into morning/afternoon, wet/dry seasons, and young/adult/old. Statistics for temporal variables (arrows) are shown. Source data are provided in the source data file.

($R^2 = 0.003$, $p = 0.22$; Fig. 3c). The microbiomes of old meerkats (>5 years old) more likely to be dominated by *Romboutsia*, and *Paeniclostridium*, than the microbiomes of adults (1–5 years old) and young meerkats (< 1 year old; Fig. 3c).

Changes to beta diversity are driven by shifts in relative abundance rather than absolute abundance. We therefore modelled genus-level dynamics in absolute abundance over daily, seasonal and lifetime scales. We first performed simple differential abundance non-parametric tests across all genera with over 15% prevalence across samples ($n = 117$) to identify genera that were differentially abundant in the morning compared to afternoon, in the dry season compared to the wet season, young meerkats versus adults, and adult meerkats versus

old meerkats (Supplementary Fig. 4). Almost all genera were significantly associated with time of day (Supplementary Fig. 4a), suggesting that diurnal oscillations are widespread across gut microbiome members. Only a few genera significantly differed between dry and wet seasons (Supplementary Fig. 4b). A small number of genera were differentially abundant in adults compared to young meerkats (Supplementary Fig. 4c), whilst none were differentially abundant in old meerkats compared to adults (Supplementary Fig. 4d).

We next focused on 16 notable genera in order to model their temporal dynamics using GAMMs whilst controlling for potentially confounding methodological variables. We focused on the most prevalent and abundant genera ($n = 13$) which all

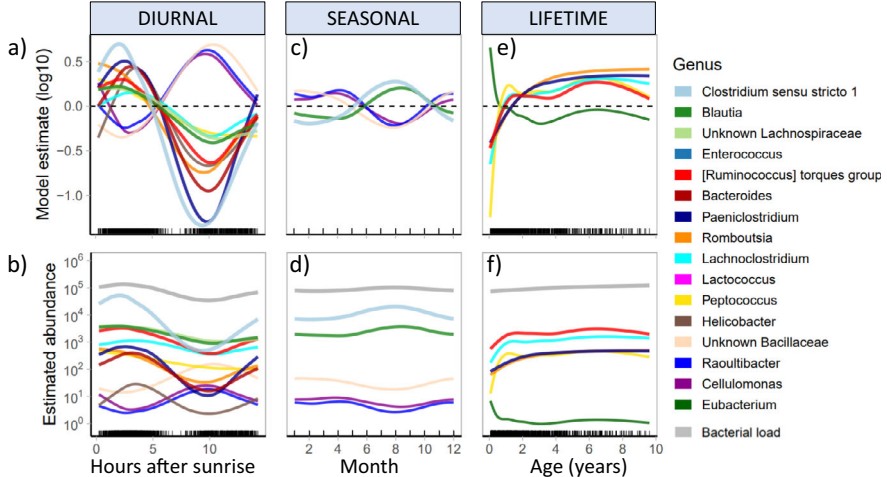

**Fig. 4 Temporal dynamics of 16 focus genera across scales.** Comparison of temporal dynamics of 16 focus genera across diurnal (**a**, **b**), seasonal (**c**, **d**), and lifetime (**e**, **f**) scales. Top panel shows GAMM abundance estimates across the three temporal scales compared to the mean, where zero (indicated by the dashed line) represents the mean (log10) abundance of each genus. Estimates have been back-transformed in the bottom panel to represent absolute abundance, and bacterial load (grey) is shown for comparison. Only genera that significantly shift across both frozen and freeze-dried samples are shown (see Supplementary Figs. 6–8 for 95% confidence intervals and trends split by storage). Source data are provided in the source data file.

had at least 60% prevalence across samples and together accounted for 75% relative abundance. However, we used the results from the differential abundance analysis to select three additional rarer genera that exhibited notable trends for additional analysis, including *Raoultibacter* (43% prevalence), and *Callulomonas* (38% prevalence). We also include a particularly rare genus, *Eubacterium* (18% prevalence), which was only present in young individuals.

Thirteen of the 16 focus genera significantly shifted over the day (Fig. 4a, b), whilst five genera fluctuated with season (Fig. 4c, d), and six were associated with age (Fig. 4e, f). Relative to their mean abundance, most genera followed the same diurnal pattern, with the highest abundance in the early morning, and decreasing until the late afternoon approximately 10 h after sunrise, before increasing slightly again prior to sunset (Fig. 4a, b). *Clostridium* exhibited the strongest diurnal oscillations, and made up a large proportion of bacterial load in the morning (Fig. 4b). *Raoultibacter, Cellulomonas,* and a Bacillaceae genus increased in the afternoon as the abundances of most genera fell, yet in general still maintained relatively low abundances (Fig. 4b). Across the year, *Clostridium* and *Blautia* increased over the dry (winter) season, whilst *Raoultibacter, Cellulomonas,* and the Bacillaceae genus increased over the wet (summer) season (Fig. 4c, d). Over meerkat life, six genera, including *Peptococcus, Paeniclostridium, Romboutsia,* and *Lachnoclostridium,* increased over the first 1–2 years of life before levelling off (Fig. 4e, f). In contrast, *Eubacterium* decreased over the first 6 months of age. There was no evidence for changes to any taxa in old age (Fig. 4c), although we note that there was a tendency for old individuals to have reduced abundance of Christensenellaceae (Supplementary Fig. 4, Supplementary Fig. 5). Trends for each genus individually, including 95% confidence internals and split by storage, are presented in Supplementary Fig. 6 (diurnal trends), Supplementary Fig. 7 (seasonal trends), and Supplementary Fig. 8 (lifetime trends).

Overall, a summary of effect sizes across diurnal, seasonal, and lifetime scales for all models presented thus far is presented in Fig. 5a, showing that diurnal oscillations are generally stronger, more prevalent, and more robust than seasonal and lifetime trends for both community and single-genus phenotypes.

**Microbiome dynamics is associated with temperature and foraging schedules.** Our third aim was to investigate the

biological mechanisms underpinning gut microbial dynamics. We were particularly interested in exploring how well climate and foraging schedules explained diurnal dynamics, which climatic variables best explain seasonal changes to the microbiome, and whether body condition and social status underpin changes to genera associated with ageing. We therefore examined the potential mechanisms underpinning temporal dynamics in five measures of bacterial load and diversity and abundance of the 16 focus genera by building GAMMs incorporating eight environmental, biological, and behavioural variables. These mechanistic variables were cumulative rainfall over the previous month, maximum and minimum temperature on the day of sample collection, the temperature at the time of sample collection, meerkat sex, body condition at the time of sampling, social status (dominant/subordinate), and the number of hours spent foraging at the time of sample collection (see Supplementary Table 1 for definitions). Foraging schedules were estimated from long-term observational data across the year, and consisted of a foraging period in the morning and a foraging period in the afternoon which shifted across the year (Supplementary Fig. 9). To summarise results, we present effect sizes of (non-linear) temporal trends with mechanistic variables excluded (blue points, Fig. 5a), and effect sizes for just mechanistic variables with temporal smooths excluded (red points, Fig. 5b), categorising associations by how robust they are to methodology.

As demonstrated in Figs. 2–4, diurnal dynamics were much stronger than seasonal and lifetime scales for both diversity measures and genus-level abundances, and were also more robust to splitting the dataset by sample storage (Fig. 5a). When considering mechanistic variables, diurnal oscillations of bacterial load, alpha and beta diversity, as well as the abundance of most genera, were largely associated with temperature and foraging schedule (Fig. 5b). Most genera, and notably *Clostridium,* increased over the foraging period, and decreased as diurnal temperatures climbed. Conversely, most diurnally oscillating taxa also increased with maximum temperature. Together, these indicate that bacterial load and abundances of *Clostridium, Paeniclostridium,* and *Romboutsia,* as well as others, peak early in the morning on hot days, likely tracking foraging schedule. Rainfall had little effect on microbiome composition, and neither did minimum temperature. Bacillaceae, *Roultibacter,* and *Cellulomonas,* in contrast, were associated with low maximum

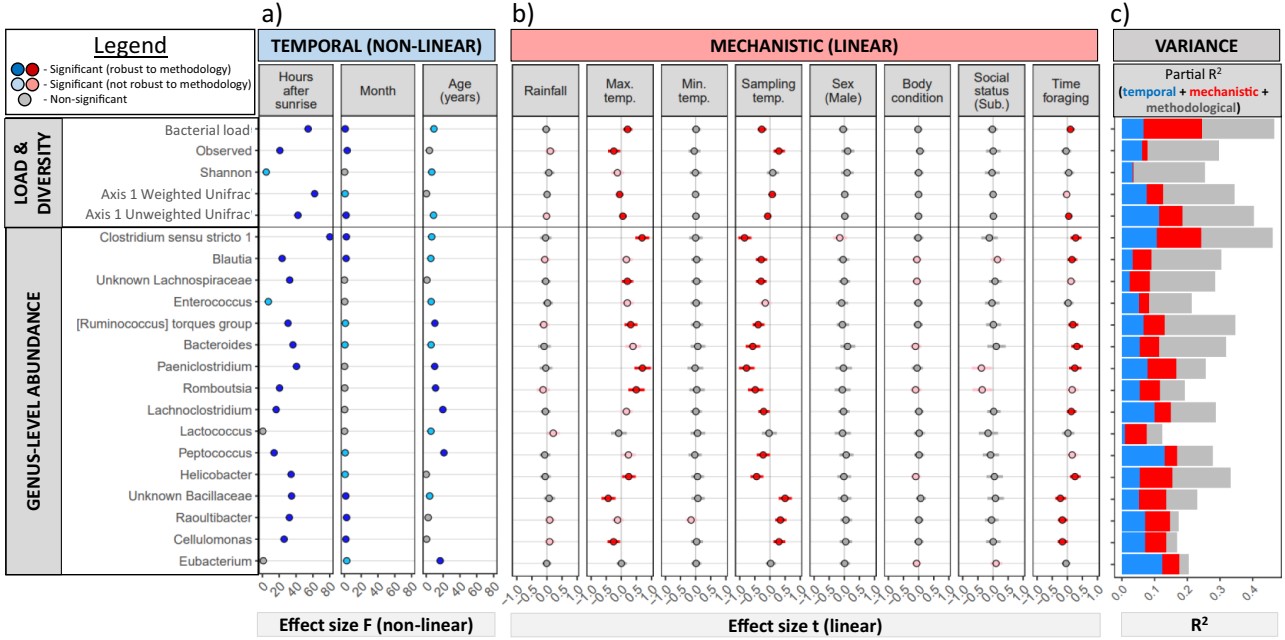

**Fig. 5 Summary of temporal (blue) and mechanistic (red) effects on the microbiome ($n = 1109$). a** Effect sizes for non-linear temporal variables on bacterial load, diversity measures, and abundances of 16 focus genera. **b** Effect sizes and 95% confidence intervals for eight environmental, biological, and behavioural variables. Dark blue/red denotes significant associations that are consistent across storage treatments, whilst light blue/red denotes significant yet inconsistent across storage. **c** Partitioning of model $R^2$ into variance explained by temporal, mechanistic, and methodological variables. Source data are provided in the source data file.

temperatures and high sampling temperature, and decreased with foraging.

We found little robust evidence for sex, body condition, or social status playing a strong role, although there were a number of weak signals across the whole dataset, albeit not replicable between frozen and freeze-dried samples. Abundances of *Paeniclostridiam* and *Romboutsia*, both of which increase with age, tended to be more abundant in dominant individuals than subordinates. Yet, these results were not replicable, potentially due to reduced statistical power in split datasets combined with the relatively few samples from dominant individuals (201 dominant vs 908 subordinate). Moreover, we found little evidence for associations between the gut microbiome and body condition at the time of sampling, although, again, a number of genera such as *Bacteroides* were weakly and negatively associated with the condition when considering all data together.

We partitioned model variance by excluding temporal and mechanistic terms from the global model that included all terms (Fig. 5c). Models explained 15–44% in variation in abundance, including methodological terms. Whilst mechanistic variables accounted for part of the variation in microbiome dynamics, a substantial proportion was only explained by temporal variables, suggesting that additional process (e.g. light-dark cycles) are at least partially responsible for diurnal oscillations. Overall, temporal and mechanistic variables accounted for approximately 5–20% in variation across measures.

**Diurnal oscillations do not decay with age.** Finally, we tested whether genus-level oscillations decrease in magnitude in older individuals by splitting samples into three age categories: samples taken from meerkats under 1 year old ($n = 385$), from adults between 1- and 5 years old ($n = 627$), and samples from old meerkats that were over 5 years of age ($n = 97$). Patterns in diurnal oscillations were very similar across all age categories, with *Clostridium sensu stricto 1* demonstrating the strongest oscillations across all groups (Fig. 6a–c). We looked in closer

detail at five genera (*Clostridium*, *Bacteroides*, *Paeniclostridium*, *Cellulomonas*, and *Raoultibacter*) that demonstrated the strongest non-linear diurnal trends and found that confidence intervals around the mean overlapped across the age categories (Fig. 6d–h). Therefore, we found no evidence for a reduction in circadian rhythms with age.

## Discussion

Understanding temporal dynamics in gut microbial communities is essential if we are to identify the mechanisms by which they shape host health and fitness. To date, many aspects of gut microbial temporal dynamics, including diurnal oscillations and changes with age, remain understudied. Our results demonstrate that gut microbial communities of wild populations can exhibit strong diurnal oscillations, and that these can dominate over seasonal or lifetime effects. Meerkat gut microbiomes exhibit a spike in bacterial load at dawn, peak in observed ASV richness at noon, and shift from a *Clostridium*-dominated community in the morning to a more diverse and low-abundance assemblage in the afternoon, distinguished by increased abundances of *Raoulti-bacter* and *Cellulomonas*. We provide evidence that these cyclical fluctuations are explained in part by temperature-constrained foraging schedules, yet patterns also suggest an equally important role of light-dark cycles and/or niche modification. Our findings are in line with those from laboratory mice and humans, which also report spikes in bacterial load when mice become active at dusk[6,19], and a peak in alpha diversity at noon in humans[7]. However, our results are intriguing in that we found strong diurnal oscillations, but weak seasonal signatures. These are likely to reflect arid-zone conditions, which are characterised by large diurnal temperature changes, paired with a largely insectivorous diet, with limited diet switching between wet and dry seasons[28].

We found that diurnal oscillations of *Clostridium* and a number of other genera were associated with foraging schedule, as well as high maximum temperature and, conversely, low temperature at the time of sampling. Whilst temperature may

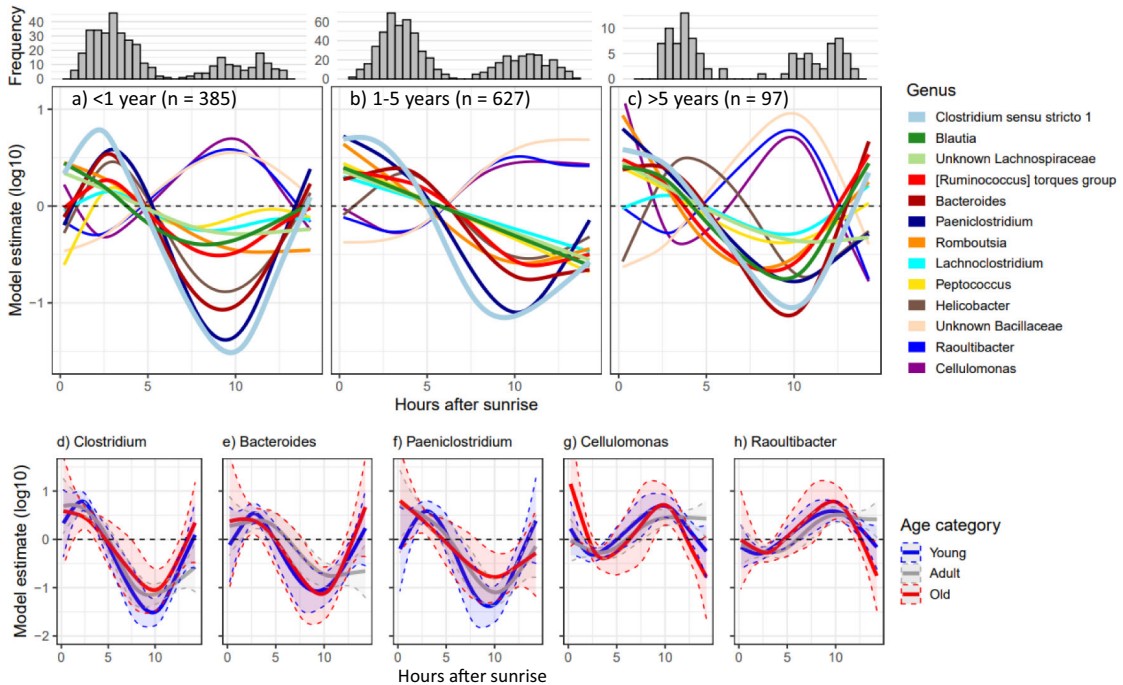

**Fig. 6 Diurnal oscillations do not decay with age. a–c** Diurnal oscillations of 16 focus genera in a) young meerkats (<1 year old; $n = 385$). **b** adult meerkats (1–5 years old; $n = 627$); and (**c**) particularly old meerkats (>5 years old; $n = 97$). Zero represents the taxa mean and sample distributions are indicated by the histograms. **d–h** Model estimates and 95% confidence intervals for (**d**) *Clostridium (sensu stricto 1)*; (**e**) *Bacteroides*; (**f**) *Paeniclostridium*, (**g**) *Cellulomonas*, and (**h**) *Raoultibacter*, split by age category (blue = young; grey = adult; red = old). Source data are provided in the source data file.

affect the gut microbiome via changes to host physiology[31–33], such associations are potentially more likely due to fine-scale tracking of meerkat foraging schedules with temperature. Our foraging schedules, however, are based on average foraging times across the year, and thus do not capture variation in foraging times on a day-to-day basis. Increases in taxa abundance with low sampling temperature and high maximum temperature are therefore likely to reflect that, on hot days, meerkats begin to forage particularly early in the morning when temperatures are cool, thereby triggering a spike in *Clostridium* and others. Nevertheless, not all temporal variation was associated with foraging schedules, potentially indicating a role for light-dark cycles and subsequent changes to hormones and immunity.

An additional mechanism maintaining diurnal oscillations may be niche modification over the day, with the spike in bacterial abundance in the morning, for instance, causing changes to gut pH and aerobic conditions[34]. This shift in the gut environment may generate favourable conditions for *Raoultibacter* and *Cellolomonas*, and supress *Clostridium* even during periods of afternoon foraging. An increase in gut oxygen levels over the day is supported by the fact that *Clostridium* is strictly anaerobic, whilst *Cellulomonas* is aerobic. Whilst we can only speculate on the function of these microbial diurnal oscillations, *Cellulomonas* degrades chitin[35], a key feature of arthropod exoskeletons, therefore increases in this genus in the afternoon may facilitate the breakdown of arthropods and other non-soluble fibres such as cellulose. Moreover, previous research has shown that *Clostridium* species generate metabolites that alter host metabolism and immunity[36–38], and that segmented filamentous bacteria (SFB), which are closely related to *Clostridium*, regulate diurnal shifts in immunity and susceptibility[16]. As such, the observed dawn spike in *Clostridium* and other Clostridiales members may be key to mediating meerkat circadian function.

In contrast to other studies on gut microbiome seasonality[8–10], we found only weak seasonal effects on the gut microbiome in

meerkats. Nevertheless, we identified seasonal shifts in a small number of taxa, including *Clostridium*, *Blautia*, and Bacillaceae. Surprisingly, these shifts were not associated with the amount of rainfall over the previous month, unlike others[39], but were instead linked to temperature and foraging. This suggests that rain-associated changes to prey diversity in the wet season may not be the major mechanism driving seasonal shifts. Instead, seasonal increases in some taxa over the dry winter may again be due to reduced foraging constrains in winter. Meerkats spend more time foraging in the middle of the day during winter[28], which may maintain higher abundances of foraging-associated taxa. It should be noted that whilst meerkats have a slight shift in diet across seasons[28], this shift is relatively small compared to omnivores that switch between food types between seasons (e.g. from fruit to leaves, or from meat to plant-based diets). Therefore, the weak seasonal effects presented here are likely to be more representative of insectivores than omnivores inhabiting highly seasonal environments.

As well as short-term dynamics, we were also particularly interested in examining lifetime processes such as microbiome development and senescence. We found little robust evidence for directional changes to bacterial load and diversity with age. Nevertheless, we do report higher variation in alpha diversity in younger meerkats than older meerkats. We also identify some genera that change over juvenile development, including a decrease in *Eubacterium* over the first year of life. This genus was also more abundant in juveniles in the Egyptian mongoose[40], and is associated with the transitional state between the infant and adult gut microbiota in humans[11,41,42]. Therefore, *Eubacterium* likely represents the weaning period, when young meerkats transition from a milk-based to an arthropod diet. Across development, *Eubacterium* was replaced by a number of adult-associated taxa, including *Lachnoclostridium*, *Ruminococcus*, *Romboutsia* and *Paeniclostridium*. These taxa stabilised at different points, with *Romboutsia* and *Paeniclostridium* stabilising at 2 years of age, when meerkat body mass

plateaus[29], whilst the others peaking at approximately 1 year of age, when meerkats reach sexual maturity. However, since these taxa were not reliably associated with body condition nor social status, the underlying reasons for their increase remain unclear.

Lastly, we found little indication of chronological senescence of the microbiome in old age, although we highlight that there was a tendency for *Christanellaceae R7 group*, a genus consistently linked to health and longevity[12,43] to decline in old meerkats. We also expected that microbial senescence might take the form of reduced microbial diurnal oscillations, since organismal senescence is characterised by a decline circadian rhythms in circulating hormones and immunity[26,44]. However, the strength of microbial diurnal oscillations was remarkably similar across age groups, although the power to detect declining rhythms is limited by smaller sample sizes for old individuals. One potential explanation for this is that whilst meerkats undergo reproductive senescence in older age, this is not paired with reduced survival rates[29], potentially due to the benefits of group living. Whilst there are few studies investigating age-related decline in wildlife gut microbiomes, our findings reflect those found in chimpanzees, which also are a group-living species and where old individuals have similar gut microbiomes to other adults[20]. This raises the question as to whether the buffering of senescence by sociality, for which there is some evidence[45], may limit microbial senescence in old age. Nevertheless, older chimpanzees in the same population have reduced diurnal hormonal cycles[26], and therefore it is conceivable that they may also exhibit diminished gut microbial circadian rhythms, yet this has not been tested. Senescence in old age is widespread in wildlife[46], and further investigation on the association between senescence and the microbiome is needed to elucidate the role of gut microbiota in host ageing.

Our study combined extensive longitudinal data and microbiome load quantification to advance our understanding of temporal dynamics in gut microbiomes. Nevertheless, it faces some study design and methodological limitations that may affect interpretations. Notably, the use of internal standards is likely prone to high technical variation, since it is challenging to accurately standardise sample weight, and subsequent technical variation can be inflated by PCR bias[47]. Our technical replication analysis confirmed that technical variation was higher for estimates of bacterial load (10%) than measures of alpha and beta diversity (~2%). Whilst this variation is non-negligible, sample ID still accounted for 90% of variation and therefore the identification of true biological associations is possible, especially with large sample sizes. We also minimise the risk of further PCR bias by controlling for sequencing depth in all analyses[47]. A perhaps more serious concern is that variation in 16 S rRNA gene copy number biases bacterial load estimates due to differences in the number copies between bacterial species. To date there is no consensus about how to control for 16S copy number in amplicon data[48], and bacterial genomes can contain between one and 21 gene copies[49,50]. As such, our estimated abundances are almost certainly over-estimates. *Clostridium* species predictably have high copy numbers (~10 copies), therefore at least part of the large spike in *Clostridium*, and reflected in bacterial load, may be an artefact of high copy number. Nevertheless, we are interested in estimating relative changes in abundance over time within communities, rather than comparing abundances amongst taxonomically different communities. Therefore, whilst the rates of change over time are not comparable between different taxa, the overall direction of change for each taxa is reliable.

Overall, the strength of diurnal oscillations identified here suggest that circadian rhythms in gut microbiomes are likely to play a major role in host biological function. Exposing the universality of gut microbial circadian rhythms among wildlife species, and the evolutionary and ecological mechanisms that underpin them, are major avenues of research in the future. The results of this study therefore form a concrete reference point from which to develop our understanding of the link between circadian rhythms in gut microbial communities and host biological function, fitness, and health.

## Methods

**Study population and study design**. We aimed to understand gut microbiome dynamics of meerkats (*Suricata suricatta*) inhabiting the Kalahari desert region in South Africa (−26.96S, 21.83E). Individuals from this population are individually marked and have been monitored almost daily since 1995 by the Kalahari Meerkat Project[27]. Faecal samples have been collected across the entire study period from almost all monitored individuals. For this study, we selected a total of 1109 samples from 235 individuals for microbiome analysis (mean = 5, min. = 1, max. = 14; Fig. 1a). These individuals were chosen either because they lived beyond the age of known biological senescence (5.5 years; [16]), or because they lived during two focus periods spanning 2–4 years approximately a decade apart. These focus periods were chosen to be able to account for environmental variables experienced by multiple study animals within each timeframe. 70% of samples were collected from meerkats belonging to eight social groups, whilst the remaining 30% were collected from across 34 different social groups. Samples were collected mostly between 6 am and 1 pm, and then again from 3 pm until 8 pm, which are the periods that meerkats are most active. Fifteen sand samples collected across the study area in June 2019 were also included to identify likely sand contaminants.

**Sample collection and storage**. Faecal samples were collected from the ground immediately after an individually marked meerkat was observed defecating. Samples were stored next to an icepack during the remaining morning or afternoon field session and were immediately frozen after return to the field station. Samples collected prior to 2008 were almost all stored frozen at −80 °C, whilst those collected after 2008, were freeze-dried for long-term storage and kept at room temperature at the Kalahari Research Centre (see Supplementary Fig. 1).

**DNA extraction with internal standard, 16 S rRNA amplification and sequencing**. Before extraction, NAP buffer was added to all faecal samples[51]. A subsample of 0.6 ± 0.05 μg (wet) was taken, and sixteen technical replicates (eight frozen, eight freeze-dried) were also subsampled at this point. 3 μl of Zymo-BIOMICS Spike-in Control I (High Microbial Load) was added to each subsample prior to DNA extraction. This internal standard consists of cells belonging to *Imtechella halotans* and *Allobacillus halotans*, two species which are rarely found in gut microbiome communities. An internal standard allows us to quantify ratios of absolute abundance by adding a known number of cells to each sample by which to normalise microbiome counts after sequencing. This method technically measures 16S copy number rather than absolute abundance, but has shown to accurately reflect variation in absolute abundances when care is taken to standardise faecal sample mass[52–55]. The bacterial genomic DNA was extracted using the NucleoSpin 96 Soil kit (Macherey-Nagel) following the manufacturer's instructions, and the hypervariable V4 region of the 16 S rRNA gene was amplified using the primer pair 515F (5-GTGCCAGCMGCCGCGGTAA-3) and 806R (5-GGAC-TACHVGGGTWTCTAAT-3). We used the Fluidigm Access Array™ for Illumina Sequencing Systems for indexing and adding Illumina adaptor sequences. After purification (NucleoMag® NGS Clean-up and Size Select, Macherey-Nagel) and quantification (QuantiFlour® dsDNA Systemt, Promega) of barcoded samples, the normalised pooled sample library was sequenced as paired-end run on Illumina MiSeq platform at the Institute of Evolutionary Ecology and Conservation Genomics, Ulm University. Samples were sequenced across four Illumina runs (MiSeq Reagent Kit v2, 500-cycles), with samples from different focus periods and storage methods distributed randomly across extraction plates and runs. Extraction and PCR negative controls were included on all runs.

**Bioinformatics and normalisation**. All sequence reads were processed using QIIME2 version 2020.2[56]. Sequences were merged, quality filtered, and chimera filtered using the DADA2 pipeline[30] to generate amplicon sequence variants (ASVs)[30,57]. Primers were trimmed and reads were truncated at 244 (forward) and 235 (reverse) base pairs. ASVs were assigned a taxonomy using SILVA version 132[58]. A tree was built using QIIME2's fragment insertion method[59], which inserts sequences into a high-quality reference phylogeny and thus provides more accurate branch lengths and tip placements than de-novo tree assembly. ASVs were filtered if they were not bacteria, not assigned to a phylum (as these are assumed to be spurious), or if they were classified as mitochondria or chloroplasts. This filtering step removed 1.7% of reads.

We aimed to remove sand contaminants that were added during faecal sample collection, yet retain sand microbes that were commonly ingested since meerkats dig for their prey and therefore take-up sand during foraging. We used the function *decontam::isContaminant*[60] using the 'prevalence' method to identify sand microbes using 15 sand samples as a reference, and to remove them from the dataset. One ASV, belonging to the genus *Geodermatophilus*, had very high

occupancy both for sand and faecal samples, therefore we retained this ASV in the dataset under the assumption that it is commonly ingested and is a real gut microbe. This filtering step removed a further 4.3% of reads. In addition, 11 low-abundance ASVs (making up 0.003% of reads) were identified with the function *decontam::isContaminant* as laboratory contaminants and removed from the dataset.

We used counts of internal reference species *Imtechella halotans* and *Allobacillus halotans* to quantify ratios of absolute abundance across samples. The two spike-in species together made up a median of 3.3% and a mean of 10% of overall reads, although a small proportion of samples had much higher percentage. *Imtechella* and *Allobacillus* counts were 99% correlated, therefore we scaled samples to *Allobacillus*. The sample scaling factor was generated by multiplying the mean read count of *Allobacillus* by its read count in each sample, and sample reads were then multiplied by the sample scaling factor to normalise the dataset. Both *Allobacillus* and *Imtechella* were then removed, and all further analysis were conducted on scaled reads. Because some samples had very high relative abundances of spike-in, we only retained samples where read depth of the true microbiome (minus the internal reference) was over 10,000 before normalization ($n = 1109$). All further analyses were conducted using normalised data.

**Estimating technical variation**. We estimated technical variation from the 16 technical replicates by calculating bacterial load, ASV richness, and four measures of beta diversity for each sample. We then estimated the variation attributed to sample ID by fitting Generalised Linear Mixed Models (GLMMs) using the *lme4::lmer* function[61] with sample ID as a random effect to each diversity measure. Variation was extracted in the form of the Intraclass Correlation Coefficient (ICC) using the *performance::icc* function[62].

**Sample metadata**. We collated temporal, biological, environmental and methodological data for each sample (Supplementary table 1). Biological data was extracted from the Kalahari Meerkat Project database. As well as the time and date the sample was collected, we included age at sampling, sex, social status (dominant/subordinate), social group and weight (in grams) at the time of sampling. We measured time of day in reference to sunrise, because this is more biologically meaningful than time of day. We calculated sunrise times per day using *suncalc::getSunlightTimes*[63]. Detailed explanations on the calculation of body condition, foraging schedule, number of hours between sample collection and freezing, and the collation of weather data is outlined in Supplementary methods 1.

**Data analysis**. We chose to model temporal dynamics with GAMMs because we reasoned temporal dynamics across scales were likely to be non-linear, and we aimed to understand fine-scale temporal trends rather than bin data into coarse categories (e.g. morning/evening, wet/dry seasons, and young/adult/old). Due to the complexity of the data and some methodological limitations (e.g. different storage methods and unequal sampling across the day) we perform a number of sensitivity analyses to assess the robustness of reported associations which are described in detail below. Overall, we only consider an association robust if it can be replicated across both frozen and freeze-dried samples, which removes weak or spurious results. All analysis was conducted in R version 3.6.2, and analyses and visualisations were performed with the key packages phyloseq[64], vegan[65], mgcv[66], and gratia (https://gavinsimpson.github.io/gratia/).

**Modelling temporal dynamics in bacterial load (Aim 1.1)**. We fitted Generalised Additive Mixed Models (GAMMs) using the *mgcv::gamm* function with a Gaussian distribution to model changes to mean log10-transformed bacterial load across the three temporal scales. Whilst count data is normally modelled with Poisson or negative binomial distributions, our abundance data spanned three orders of magnitude, and models using Poisson or zero-inflated negative binomial distributions did a poorer job of modelling abundance counts than transformed counts with a Gaussian distribution. We included hours after sunrise, month, age at sampling, and sequencing depth as continuous smooth terms; individual ID and social group as smoothed random effects; and storage, sequencing run, and field time as linear parametric terms. All smoothed terms except month were fitted with cubic regression splines (bs = "cr"), whilst month was fitted with a cyclic cubic regression spline (bs = "cc"), because seasonal changes are cyclical (i.e. January comes after December). Cubic regression splines calculate smoothing knots based on data density (rather than distributing them equally along a gradient), and therefore periods of missing data, e.g. during the middle of the day, do not contain knots nor generate erratic trends. We included individual ID and social group as random effects by fitting random spline (bs = "re") to these terms. Because temporal data can be temporally auto-correlated (i.e. samples collected at the same time are not temporally independent), we added an nested autoregressive model to account for temporally correlated errors within the GAMM. We nested the autoregressive term within sample year, because model comparison indicated this marginally improved model fit. However, across models presented in the study, choice of smoothing method and inclusion of an autocorrelation term made little difference to results.

We validated our results in a number of ways. First, we fitted smoothed trends separately to frozen (samples collected prior to 2008) and freeze-dried (mostly collected after 2008) samples, using *mgcv*'s 'by' argument. We present results in the main text and only consider associations robust if they are replicated across both storage types. Secondly, because samples were collected unequally across the day, with few samples in the middle of the day, we randomly subsampled 20 samples per hour interval (minus two hours at noon that had fewer than 20 samples and were therefore excluded) and re-ran models on the reduced dataset, and found results were robust to sampling distribution (Supplementary Fig. 10a). Thirdly, because methodological variables appeared to have large effects on results, we repeated analyses without any methodical variables, and found temporal dynamics were largely unaffected (Supplementary Fig. 10b). This indicates that including methodological variables increases explanatory power of models but not do not overly bias estimates. Fourthly, we cross-validated our models by building them on a random subset of 70% of our dataset, then applying the model to the other 30% untrained data, and repeated this 100 times. Average $R^2$ of cross-validation models was 43% (compared to 47% for presented model), indicating our models were not over-fitted to trained data.

**Modelling temporal dynamics in alpha diversity (aim 1.2)**. ASV richness and Shannon diversity were calculated on counts normalised to the internal standard using the function *phyloseq::estimate_richness*. We focus on observed ASV richness as a measure of alpha diversity, yet also present results of Shannon diversity, which weights for abundance and therefore also represents the evenness of the community. We modelled temporal dynamics of untransformed ASV richness using the same GAM model structure as described above for bacterial load.

We performed the same sensitivity analyses as described above for the model on bacterial load, including running analyses separately on frozen and freeze-dried samples, presented in the main text. Secondly, we randomly subsampled 20 samples per hour interval (minus two hours at noon) and reran the GAMM on the reduced dataset, which produced similar results to those presented (Supplementary Fig. 11a). We also removed methodological variables and reran the model, which made little difference to model estimates (Supplementary Fig. 11b). Finally, we validated the model by splitting the dataset into training and test sets 100 times. The model explained on average 24% of variation in the untrained data (in comparison to the 29% reported), indicating model predictions were likely over-fitted and true explanatory power was closer to ~24%.

**Modelling beta diversity (aim 1.3)**. To calculate and visualise beta diversity, we excluded ASVs with total counts under 50 reads (after scaling to the internal standard). This was to ensure convergence of all ordinations, and resulted in the retention of 4,666 out of the original 26,122 ASVs, yet excluded only 0.7% of total variation. We ordinated taxa composition using Multi-Dimensional Scaling (MDS) analysis of Weighted Unifrac distances, applying counts normalised to the internal standard. We did not use non-metric MDS (NMDS) ordinations since variation was extremely high and ordinations did not converge. Whilst we focus on Weighted Unifrac, we also present results for Unweighted Unifrac, to compare weighted and unweighted measures. We statistically tested the marginal effect of temporal and methodological variables on overall beta diversity distance with PERMANOVAS using the *vegan::adonis2* function. We statistically tested for differences in centroids across axes 1 and 2 (Fig. 3b) and 3 and 4 (Fig. 3c) by using the *vegan::envfit* function, including methodological variables, and with 999 permutations. The *envfit* function uses linear model permutations to map variables onto an ordination. To summarise temporal dynamics in beta diversity and to easily compare them to trends in load and alpha diversity, we also modelled axis 1 of both Weighted and Unweighted Unifrac ordination using GAMMs, controlling for methodological variables (Supplementary Fig. 3 and Fig. 5). We fit these GAMMs to frozen and freeze-dried samples separately to assess robustness (Supplementary Fig. 3d, e).

**Genus-level analyses (aim 2)**. To understand which genera were associated with which temporal scale, we merged reads by their genus using the *phyloseq::tax_glom* function. We focused on 13 of the most common genera with over 60% prevalence, which together made up 75% relative abundance. However, we also wished to detect any meaningful dynamics in rarer taxa. We therefore ran a non-parametric differential abundance analysis on all genera with over 15% prevalence to assess any differences in taxa between morning (< 7 h after sunrise; $n = 743$) and afternoon (> 7 h after sunrise; $n = 366$), dry (May–September; $n = 418$) and wet seasons (October–April; $n = 691$), and young (< 1 year; $n = 385$) and old (> 5 years; $n = 97$) meerkats. To ensure sample sizes were not highly unequal between groups, we compared samples from both young and old meerkats to a subset of samples from meerkats between 2 and 4 years old ($n = 236$). Because most taxa are zero inflated, we used non-parametric Wilcoxon tests with Bonferroni adjusted p-values for multiple testing to test for significant differences. We used the differential abundance analysis to add four genera that exhibited notable temporal dynamics to our list of focus taxa.

We modelled dynamics of each focus genus over time by fitting a GAMM to log10-transformed abundance, adding a pseudo-count of one to zeros. GAMMs were built using an identical model structure to that described for bacterial load,

and therefore included only temporal and methodological terms, and included individual ID and social group as random effects. For rarer taxa which were zero inflated due to low prevalence (*Cellulomonas*, *Raoultibacter*, and *Eubacterium*) we also modelled abundance using the negative binomial family with zero inflation. Since these presented overall similar trends, we used the same gaussian model structure for all genera so that effect sizes are comparable. To ensure robustness, we present in the main text only trends that were significant when frozen and freeze-dried samples were analysed separately, but present all trends by storage in supplementary materials.

**Investigating underlying mechanisms (aim 3)**. We investigated underlying mechanisms by additionally including eight biological and environmental terms outlined in Supplementary Table 1 to GAMMs predicting bacterial load, observed ASV richness, Shannon diversity, ordination axes one and two of weighted and unweighted Unifrac beta diversity ordinations, and the abundances of the 16 focus genera. We first present effect sizes for temporal trends minus mechanistic variables, so that temporal dynamics of each diversity metric and genera can be easily identified. We then present effect sizes for models with mechanistic but not temporal variables included. We removed temporal non-linear variables from these models because controlling for them reduced statistical power to detect associations with underlying mechanistic variables, and to remove issues with co-correlation between time of day and temperature variables. We partitioned model $R^2$ into temporal, mechanistic, and methodological variation by removing either temporal or mechanistic variables or both from the global model (including all terms) and recording the drop in explanatory power. All models were repeated on frozen and freeze-dried samples separately and robust associations are presented.

A potential confounding issue was the co-correlation between many of the climatic variables included in the model (cumulative rainfall, minimum temperature, maximum temperature, and temperate at the time of sampling). To test whether co-correlation affected our interpretation of results, we replaced these variables with uncorrelated principal components from a PCA analysis. Overall, PCs representing maximum temperature and temperature at sampling demonstrated almost identical trends to models with the untransformed data (Supplementary Fig. 12), and rainfall remained overall unimportant. Because principal components are harder to interpret and results supported models with untransformed climate data, we present models with the untransformed data. In addition, we tested for seasonal interactions by fitting smoothing splines by season, and found that associations presented in Fig. 5 were similar across wet and dry seasons.

**Testing interactions with age (aim 4)**. To test whether diurnal oscillations per genus decayed with age, we categorised samples by age category (< 1 year; 1–5 years; > 5 years) and rebuilt GAMMs per genus, fitting diurnal smooths by age category using *mgcv*'s 'by' argument. We assessed whether diurnal oscillations were comparable across age groups by comparing 95% confidence intervals for five key genera that demonstrated the largest diurnal fluctuations.

**Pilot study to test the effect of storage**. Whilst the effects of storage can be accounted for statistically, we wanted to confirm experimentally that the two storage methods used here do not overly affect the bacterial composition. We experimentally tested the effect of freezing versus freeze-drying on overall bacterial community composition by collecting fresh faecal samples from nine different captive meerkats housed at the University of Zurich. Faecal samples were frozen immediately on collection. A subsample of the sample was then freeze-dried, whilst another subsample remained frozen at −80 °C for one week. DNA was extracted and processed following the same protocols as described above, with the exception that an internal standard was not added to samples. To analyse the effect of storage on these samples, samples were normalised by rarefaction and we performed a marginal PERMANOVA on a Weighted Unifrac distance matrix, including sample ID and storage as terms.

## Data availability
All sequences and processed data used in this study are available to download from Zenodo[67]. Sequences are additionally stored under NCBI BioProject PRJNA764180. Source data are provided with this paper.

## Code availability
All code used in this study and Rmarkdown reports are available to download from Zenodo[67].

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

## Acknowledgements
The authors are grateful to the Kalahari Research Trust and the Kalahari Meerkat Project for access to facilities and habituated animals in the Kuruman River Reserve, South Africa. This paper has relied on records of individual identities and/or life histories maintained by the Kalahari Meerkat Project and collected by scientists and volunteers. We thank the Northern Cape Conservation Service for permission to conduct fieldwork, and the South African Weather Service (SAWS) for providing weather data. We thank Ben Danzer for facilitating sample collation and storage and Ulrike Stehle for contributing to lab work. Funding: The Kalahari Meerkat Project is supported by the European Research Council (Research Grant Nos 294494 and 742808 to T.H. Clutton-Brock since 1/7/2012), the Human Frontier Science Program (funding reference RGP0051/2017), the University of Zurich, and the Mammal Research Institute at the University of Pretoria, South Africa. The study is funded by the German Research Foundation to S. Sommer (DFG SO 428/15-1). Ethics: All research for this study was conducted with permission of the ethical committee of Pretoria University and the Northern Cape Conservation Service, South Africa (Permit number: EC031-13). All the methods were carried out following the approved guidelines. Author contributions: AR designed study, performed data analysis, and wrote the manuscript. KW performed lab work. MM and TCB initiated and supervised the long-term meerkat project and data collection; SS initiated and supervised the study, acquired funding for microbiome investigation, and performed project administration. All authors revised manuscript drafts and approved the final version.

## Funding

## Competing interests
The authors declare no competing interests.

## Additional information

**Peer review information** *Nature Communications* thanks Kevin Theis and the other anonymous reviewer(s) for their contribution to the peer review this work. Peer reviewer reports are available.

