## [Peer Review File · Nature Communications]

Reviewers' Comments:

Reviewer #1:

Remarks to the Author:

This study aims to assess temporal effects (diurnal, seasonal, lifelong) on the wild meerkat gut microbiome in terms of alpha diversity, beta diversity, and bacterial load. The study presents an impressive sample set spanning many individuals across several decades, assesses a wide variety of biological and environmental variables on gut microbiome variation, and concludes with interesting findings that diurnal factors exert stronger influences on the microbiome than seasonal or lifelong impacts, which has not been shown in a wild system. The study is well-written but would benefit from several additions, both statistical and discussion based. Larger changes are listed first followed by smaller changes for each section.

Larger changes

- Only one metric for alpha and beta diversity each are presented to assess the impact of temporal, biological, and environmental variables on the gut microbiome. It is more typical to present multiple metrics. Is there a reason why only these were chosen? I think it would be especially important to include a second metric for beta diversity. Only weighted UniFrac was used which is based on relative abundance of ASVs. However, there appear to be large differences in ASV richness across groups, and thus a metric assessing presence/absence of ASVs may be beneficial as the authors mention they are only explaining a small amount of overall variation (line 156).

- As sample storage method can have significant impacts on microbial communities, I appreciate the inclusion of data aggregated by storage method in figures S5-6. However, the authors mention in the methods section that trends were largely the same across storage groups and thus, storage should not have impacted results (line 504). However, especially in figures S5 C and F the trends for freeze dried vs frozen samples look almost opposite one another. I think there should be some mention of this caveat in the results or discussion.

- Some additional interpretation of the main study results is needed in the discussion. For example, is there any speculation as to what biological mechanism may drive daily oscillations in gut microbiome load/diversity based on light/dark cycles? Clostridium seemed to be a central driver of several results—what might be the significance of this group and why might it have been so plastic? Different trends for ASV richness and bacterial loads were discussed on the basis of seasonality in the discussion (paragraph starting line 300). However, these differences also appeared on a diurnal scale which wasn't discussed. Why might these trends be different (e.g. is bacterial load driven by one dominant ASV that reduced richness?)

- There appeared to be little sampling during the mid-day period (e.g. Figs 2b, 4a). Why was this and how might this impact results? This should be discussed a bit.

Smaller changes

Introduction

- Line 51: There seems to be a typo here. May-September is listed as both summer and winter months

- Line 57: Different storage methods should be added here, as not all samples were frozen afterwards?

Results

- Line 98: change "genera" to "genus"

- Line 159: Was this 62% of the 34% of overall variation explained above? Or 62% of overall variation?

- Line 183: this header indicates results regarding clostridium, but I don't think there is any mention of clostridium in this section? Consider rephrasing

- Line 202: please explain in a bit more detail how genera were identified as core. In most cases, they are taxa present in a certain number of samples. Is that what is meant by 80% prevalence here, or does this refer to relative abundance of the taxa? Is the N here the number of genera or number of samples? A list of the taxa somewhere would be helpful

- Lines 206-211: a supplemental table may be helpful to show exactly which genera were influenced by which variables, unless they are all listed in figure 6? Maybe that figure should be cited here.
- Line 227: Were only the genera listed as core used for this analysis?

Methods

- Line 360: some additional information is needed about this storage experiment. How were the samples handled after collection? Were the samples split into storage treatments from single individuals or different individuals were used for each type? etc.
- Line 363: what statistical methods were used to compare beta and alpha diversity between these groups?
- Line 392: what version of QIIME2 was used?
- Line 397: change "that" to "than"
- Line 421: what units was weight measured in?
- Overall for beta and alpha diversity metrics, was rarefaction used to standardize sequencing depth?
- Line 518: why was MDS used as opposed to NMDS which appears to be more common for these analyses?

Figures/Tables

- Fig 2a is really difficult to glean information from due to the number of samples and microbial taxa. It may help to distill the number of taxa displayed to smaller number (e.g. top 5-10 most abundant)
- Fig 2 d-e: what cutoffs were used to delineate groups (e.g. what hours were used for morning/afternoon and months for wet/dry season). This is explained for age but not these two.
- Fig 2 legend: change GAM to GAMM?
- Figure 3a: it is unclear to me what joint vs. independent represent. This should be explained in more detail in the text or legend.
- Table S3 caption: change "statics" to "statistics"?

Reviewer #2:

Remarks to the Author:

Risley et al. extensively document microbiome variation in wild meerkats, isolating dynamics associated with circadian rhythms, seasonality, and host demography. The authors leverage an extensive sample and metadata set to answer questions previously untackled in microbiome analyses, and they use appropriate mixed effects models to do so. The paper provides a unique perspective on gut microbiome dynamics in a wild population and is a significant contribution to the field.

I have identified four larger issues that should be addressed before publication however and also provide a list of smaller concerns.

Larger Issues:

Model choice—The authors do a laudable job of describing their GAMM fitting in the methods, but their use of hierarchical variance partitioning models is much less well motivated. The R package used is not specified nor are the validation tests. Moreover, the authors fail to explain (i) why the hierarchical variance partitioning model approach was used to analyze the biological predictors of the first four major principal components of the MDS ordination and (ii) why it wasn't used for analyses of biological predictor role for other dependent variables. Furthermore, why analyze biological fit on 4 MDS axes rather than just modeling fit of overall dissimilarity with PERMANOVA?

Abundance patterns— The fact that all but one major genera exhibit the same diurnal shifts despite being associated with different biological and environmental variables is confusing to me. To what extent may these just be reflecting the overall trend in load? The strong effect of sequencing depth for all genera except *Geodermatophilus* seems to indicate this may be at play. Are similar shifts observed when analyzing relative abundance? (I'm not advocating including relative abundance analyses in the published paper, just using them as a tool to better understand

why absolute abundance patterns appear as they do.)

The extreme response exhibited by *Clostridium* may in part reflect the fact that it has much higher (2-5x) 16S copy number than the other genera analyzed (for which genomes are available and copy number estimates are published in rrnDB). While the authors note in the methods that the internal standard technique directly assesses copy number but generally reflects absolute abundance trends, they don't discuss how large differences in copy number will impact their interpretation.

Diurnal time series—For totally understandable methodological/behavioral reasons, the authors were unable to sample evenly across the course of the day. However, they fit their models across the entire day even though between ~5 and ~10 hours after sunrise their sampling is very sparse. The confidence intervals are somewhat larger there, but I'm not sure you should be fitting through at all. Certainly more discussion of this limitation is necessary.

It's unclear to me as a non-meerkat specialist how frequently an individual defecates so whether there could potentially be samples in the middle of the day if one had access to burrows or even at multiple times during the day for one host. It is hard to tell from Figure 1 whether there are individuals who were ever sampled multiple times on the same day even during the two dense sampling campaigns. If there are, would it be possible to provide supplemental figures plotting their observed richness or load over such a day? This would be especially helpful for estimating how representative the overall fit is for individuals given that there is such variance at any given time of sampling.

Dominant genera—The specification of "dominant genera" for the enterotype analyses (Figure 3) needs to be better defined in the results and the methods. Is it just the most abundant genus? Why aren't these the genera of focus for the core genera analyses if they are what distinguishes between communities?

At the very least the color schemes should be consistent between 3b and 4e-h so the reader can more easily tie them together.

Smaller Concerns:

-line 39 It is unclear what the definition of "biological systems" is in this context. Do you mean non-microbiome host physiological programs?

-line 47-50 What are animals doing when they aren't foraging?

-line 65 Specify you mean the abundance of each of the 15 genera, not the 15 together.

-line 68 What kind of models?

-lines 98, 99 Specify if 60% and 30% are means and provide standard deviation or confidence intervals for the summary statistic.

-The presentation of the GAMM model fits is a bit hard to follow in the text (e.g. lines 106-116; 144-156). Is there a way to report statistical significance for relevant variables in the text rather than just directing to a supplementary table?

-line 469-470 You're testing underlying mechanism not necessarily "identifying" them since you don't have all possible relevant mechanistic variables included in your analyses.

-line 509-512 What would've been evidence for more than marginal overfitting? Isn't a 1/4 drop in variance explained a lot?

-line 522-528 You could analyze with `adonis2` function in `vegan`, rather than `adonis` function, to capitalize on a marginal sums of squares approach (`by="margin"`) and thus not be subject to the same sequential effects.

Figures

-Please provide supplementary figures with histograms of your sample set. For example, what is the frequency of samples for time of day (maybe hourly bins), months, and age? And what is the

frequency of samples for time of day by age groups (underlying the figure 5 analysis)? It's not possible to back this out of Figure 1 or the highly dense observed data plots (like 2b-d), but such information is useful for understanding how robust the patterns are.

-I find figure 2a really hard to parse, especially since the x-axis is not plotted as a continuous axis. Either fewer families need to be plotted and/or averages for ordinal time points need to be plotted. The current version could be included in the supplement if you think it is necessary, but it is too busy to read as is. Also, 2A should not be cited after the sentence "The most abundant genera across samples was *Clostridium sensu stricto* 1, an anaerobe that made up 30% of reads, and which was more abundant in the morning than the afternoon and evening" since there are no genera results in it.

-"Dominant genera" needs to be defined in the legend of Figure 3. Is "other" any other genera or a group of non-specified but not exhaustive genera?

-Figure 4d x and y axes should match, the axes should not have decimal places in the superscript, and the blue line should be defined (is it a fit line or 1:1)?

-Why are 4h-j not also on a log scale?

-Can you provide the overall load oscillations for each age group in Figure 5? If differences in overall abundance are underlying much of the genera oscillations (see "Abundance Patterns" above) we'd expect to see a similar overall abundance curve for each age group. But if the overall signal varies while the genera patterns are consistent, that supports the idea that genera behave somewhat independently.

-Figure 6 in general is an excellent way of summarizing some very complex model results. It would be helpful though to have a legend on the plot specifying what dot size indicates.

Reviewer #3:

Remarks to the Author:

The manuscript titled "Diurnal oscillations in gut microbiome load and composition eclipse seasonal and lifetime dynamics in wild meerkats, *Suricata suricatta*" aims to quantify and compare the meerkat gut microbiome across diurnal, seasonal, and lifetime cycles.

The general approach of the study was to characterize the bacterial load and structure of the meerkat gut microbiome using 1027 samples collected in the Kalahari across 20 years of morning and evening behavioral observations of individually known meerkats. Longitudinal sampling was available for 168 meerkats. The gut microbiome was characterized via 16S rRNA gene sequencing. Bacterial load was estimated by scaling reads to internal standards (commercially available) that were introduced prior to DNA extraction. Alpha and beta analyses were performed.

The key finding of the study was that most common bacterial genera exhibited diurnal oscillations in bacterial load. These oscillations were driven by changes to *Clostridium sensu stricto* 1, and were better explained by light-dark cycles than foraging schedule. Diurnal cycling of the microbiome did not decay with age.

The conclusion of the study was that diurnal oscillations shape gut microbiome load and structure in wild meerkats.

Comments/Concerns

This is a high-quality and well written study addressing circadian rhythms of the gut microbiome in a wild population across 20 years of intensive sampling and behavioral observation. The breadth, scope, and detail of the metadata available for the samples is impressive. Commendably, the authors provide all code in an Rmarkdown file. Extraction and PCR negative controls were included on all sequencing runs.

I initially had two concerns, however, both are ultimately addressed in the study.

1) The first potential concern was that fecal samples collected prior to 2008 were frozen at -80°C, while those collected after 2008 were freeze-dried and stored at room temperature. However, the study includes an analysis showing that biological variation exceeds technical variation potentially introduced by these differences in storage method.

2) The second potential concern was that bacterial load was estimated using ZymoBIOMICS Spike-in Controls. I have never seen this approach for quantifying bacterial load and was instead expecting quantitative real-time PCR. However, after reading the product literature and related manuscripts, I appreciate the approach. As noted below, I would however recommend that the approach be described in greater detail in the Methods, as it is novel.

I therefore have only minor suggestions.

Minor comments/edits:

Additional proof-reading is recommended prior to publication (e.g., plural/singular).

Abstract

N/A

Introduction

It would be valuable to present what is known of the mongoose gut microbiome from prior studies in the Introduction. At present, there is no indication that mongoose gut microbiome has been characterized at all. If it has not, make this clear.

Materials and Methods

Line 379 – Did the kit extract all genomic DNA, including that of the host, or only of the bacteria?

Lines 388-390 – Were the DNA extractions also performed in a randomized manner?

Line 393 – What parameters were used for the DADA2 pipeline? They were not included in the Rmarkdown report.

Line 402 – What functions/parameters were used for Decontam? They were not included in the Rmarkdown report.

Line 408 – By what manner were these ASVs identified as laboratory contaminants? Given their rarity, they would not affect any study outcomes, but given the careful methodology used in the study, stating these criteria would benefit others in conducting similar studies.

Line 413 – Please explain how samples were scaled to *Allobacillus*. This is a new technique, which warrants further explanation.

Lines 424-426 – Explain how weights were collected in a near daily manner, or include a reference to prior descriptions of the process.

Results

Figure 2 – There are two “d” panels in the figure.

Discussion & Conclusions

A section on the strengths and limitations of the study would be valuable.

Other

It appears that Ben Dantzer’s name is misspelled in the Acknowledgements, unless that is not who is being referred to.

Kevin R. Theis

RESPONSE TO REVIEWERS

Summary of changes for reviewers

We would like to thank the three reviewers for taking the time to review the manuscript and for their constructive criticism. We have taken all suggestions on board in our revised paper and believe the manuscript is now considerably improved. As requested, we have made major alterations to our paper. Because some of the points brought up by reviewers as ‘discussion points’ lead us to add more data and reanalyse the data more vigorously, there has been some analytical changes that warrant an overall summary of the changes to the manuscript:

1. All reviewers thought there should be more discussion and context. We have expanded the introduction to include two paragraphs on what we know about the temporal processes we are interested in, specifically circadian rhythms and development and senescence of the gut microbiome. We have also expanded the discussion to more fully interpret our results, as well as a paragraph on the methodological limitations.
2. One reviewer requested additional diversity metrics be added to the analysis – we have added these. We added Shannon diversity as another measure of alpha diversity, and Unweighted Unifrac as an additional, unweighted, measure of beta diversity.
3. Two reviewers were concerned about the lack of data in the middle of the day and how this affected models. We agree that this situation was not ideal, so we sequenced 80 more samples that were collected during the middle of day (between 12-4pm), or very early in the morning (~6am) or late in the day (~8pm). This does not close the noon gap, but it narrows the gap from about 5 hours to 3 hours. We also go into more detail on the new sensitivity analyses we apply (random sub-sampling across the day) and why this gap does not effects estimates from other parts of the day.
4. Because we did another sequencing run, we also took this opportunity to do a small technical replication test, since as reviewer 3 brought up, the use of spike-ins/internal references is still a relatively novel method, and there is little information on how reliable it is. Sample ID counted for 90% of variation in estimated bacterial load, and therefore technical variation was 10% (technical variation of alpha and beta diversity was much lower at ~1-2%). This is relatively high, but still good enough to work with and identify biological trends. All reviewers brought up the reliability of the methods, therefore to be completely transparent we have now added a ‘methods validation’ section at the start of the results, which tackles the effect of storage on the microbiome, and the technical variation. Readers now do not need to dig into the methods to find this information.
5. Two reviewers wanted more discussion on the effect of frozen/freezefried samples on results. Possibly the largest change in terms of the interpretation of our results comes from the fact we now only consider an association robust if it exhibits the same significant trends both in freezedried and frozen samples. We believe this makes the results and conclusions much clearer, because it weeds out any weak associations that are significant just because of the large sample size. During this process, we realised that our models containing all the mechanistic variables (the ‘full models’) shouldn’t have the non-linear terms included, because these correlate substantially with some of the fixed terms in the model, and the resulting associations were not robust when we split the

43 dataset by storage. Our largest analytic change, therefore, is to rerun these models (that
include all the climate/biological variables) without temporal non-linear terms, whilst
thoroughly taking into account co-correlation. For example, we now only consider one
foraging variable instead of two due to correlation issues, and also test for the effect of
co-correlation between climate variables. This has altered our interpretations of the
mechanisms, with temperature-constrained foraging patterns becoming much more
important, which we believe makes sense with what we know of the gut microbiome. We
emphasize though that the extra data and reanalysis does not alter our overall conclusions.

6. Focus genera: There was some confusion about the selection of genera modelled and the
colour scheme of figures, since different genera were presented in different analyses
using different colours. To be more consistent, we have tested for temporal dynamics
across a larger suite of genera, but still focused in on the 16 most important. These 16
genera have the same colour scheme across all figures, and we more fully justify their
inclusion.

7. Lastly, we have slightly reorganised the results section, based on a new set of four clear
aims that we added at the end of the introduction. For example, we now focus on the
mechanistic effects in one section, whilst previously this was spread out over the
manuscript.

We have responded to each comment in detail below. We have highlighted revisions in our
revised manuscript by colouring new or changed in sections in blue.

**Reviewer #1 (Remarks to the Author):**

This study aims to assess temporal effects (diurnal, seasonal, lifelong) on the wild meerkat
gut microbiome in terms of alpha diversity, beta diversity, and bacterial load. The study
presents an impressive sample set spanning many individuals across several decades, assesses
a wide variety of biological and environmental variables on gut microbiome variation, and
concludes with interesting findings that diurnal factors exert stronger influences on the
microbiome than seasonal or lifelong impacts, which has not been shown in a wild system.
The study is well-written but would benefit from several additions, both statistical and
discussion based. Larger changes are listed first followed by smaller changes for each
section.

Thank you for this positive assessment for our work.

Larger changes

- Only one metric for alpha and beta diversity each are presented to assess the impact of
temporal, biological, and environmental variables on the gut microbiome. It is more typical to
present multiple metrics. Is there are a reason why only these were chosen? I think it would
be especially important to include a second metric for beta diversity. Only weighted UniFrac
was used which is based on relative abundance of ASVs. However, there appear to be large
differences in ASV richness across groups, and thus a metric assessing presence/absence of

ASVs may be beneficial as the authors mention they are only explaining a small amount of
overall variation (line 156).

In response we have now added Shannon diversity and Unweighted Unifrac to analyses. We
agree it is the norm for wildlife microbiome studies to present more than one diversity metric,
and our decision to present just one metric of alpha and beta diversity was based on space
alone, since we present rather a lot of information.

We have outlined the results of these analyses in the results section (L168-170), and methods
section (L 570), and included summary stats for them in Figure 5. In summary, Shannon
diversity only shows very weak temporal trends, yet unweighted Unifrac shows similar
patterns to Weighted Unifrac.

- As sample storage method can have significant impacts on microbial communities, I
appreciate the inclusion of data aggregated by storage method in figures S5-6. However, the
authors mention in the methods section that trends were largely the same across storage
groups and thus, storage should not have impacted results (line 504). However, especially in
figures S5 C and F the trends for freeze dried vs frozen samples look almost opposite one
another. I think there should be some mention of this caveat in the results or discussion.

As outlined in point 5 of the summary above, we now only consider associations robust if
they hold up both for frozen and freeze-dried samples, even if the association is significant
overall. This is because many associations can be significant if the sample size is high, yet do
not necessarily represent meaningful relationships. The associations that were inconsistent
across frozen and freeze-dried samples (e.g. bacterial load as brought up by the reviewer) were
those that tended to be weak overall, and therefore prone to being inconsistent. We now
present all trends split by storage (Figs S3, S6, S7, and S8), and colour associations by how
robust they are in Figure 5. In Figure 5, it is clear that strong associations are almost always
robust across the two storage types, whilst weak associations tend not to be robust. We state
more clearly now when presenting each result about whether we consider it robust or not.
For example, we have altered the bacterial load results to:

L145: “Mean bacterial load underwent the largest shifts across the day, in comparison to
seasonal and lifetime scales, which were both much weaker (Hours after sunrise: $F = 54.4$, p
< 0.0001 ; Month: $F = 1.1$, $p = 0.007$; Age: $F = 9.1$, $p = 0.003$; model $R^2 = 0.47$; Table S2).
Bacterial load tended to be highest early in the morning (Fig. 2a), and fluctuated only weakly
with season (Fig. 2b) and age (Fig. 2c). Whilst seasonal and lifetime shifts in bacterial load
were weak but significant across the full dataset, they were not replicable across both frozen
and freeze-dried samples (Fig S3a).”

- Some additional interpretation of the main study results is needed in the discussion. For
example, is there any speculation as to what biological mechanism may drive daily
oscillations in gut microbiome load/diversity based on light/dark cycles?

Given our new analysis of the mechanisms, we believe temperature-constrained foraging
schedules are at least partly to explain for diurnal oscillations. However, we wanted to
quantify how much variation could be attributed to foraging schedules, and how much was
explained solely by time of day (which suggests regulation by light-dark cycles and host
circadian rhythms). We therefore partition this variation in Figure 5c. The results are in line
with findings from mouse studies, which show both feeding schedule and light-dark cycles
govern microbial oscillations. Nevertheless, we suspect diurnal oscillations in this species
might be stronger than usual, due to arid conditions which are characterized by large
temperature differentials across the day, and which strongly shape foraging schedules. We
have now expanded our discussion on these points (L320-393, four paragraphs that cover
interpretation of diurnal, seasonal, and lifetime dynamics).

*Clostridium* seemed to be a central driver of several results—what might be the significance
of this group and why might it have been so plastic?

*Clostridium* has been implicated in a number of lab mouse studies on microbial circadian
rhythms (which we now outline in the introduction in L41-53). Therefore, we know already
that this genus is highly dynamic, especially over the daily time scales. Whilst we can only
speculate on its function, *Clostridium sporogenes* is known to generate metabolites that
mediate host metabolism and immunity, therefore we believe something similar might be
going on here. Moreover, gut conditions may also play a role: *Clostridium* is strictly
anaerobic, whilst at least one taxa that increases in the afternoon (*Cellulomonas*) is strictly
aerobic. Therefore, there is evidence that there is a change in oxygen levels over the day that
is likely to maintain diurnal oscillations. We now touch on this in our discussion in L332-
343.

“An additional mechanism maintaining diurnal oscillations may be niche modification over
the day, with the spike in bacterial abundance in the morning, for instance, causing changes
to gut pH and aerobic conditions³³. This shift in gut environment may generate favourable
conditions for *Raoultibacter* and *Cellulomonas*, and suppress *Clostridium* even during periods
of afternoon foraging. A shift in gut oxygen levels over the day is supported by the fact that
*Clostridium* is strictly anaerobic, whilst *Cellulomonas* is aerobic. Whilst we can only
speculate on the function of these microbial diurnal oscillations, *Cellulomonas* degrades
chitin³⁵, a key feature of arthropod exoskeletons, therefore increases in this genus in the
afternoon therefore facilitate the breakdown of arthropods and other non-soluble fibres such
as cellulose. Moreover, previous research has shown that *Clostridium* species generate
metabolites that alter host metabolism and immunity³⁴, suggesting that the dawn spike in this
genus may be key to mediating meerkat circadian function.”

Different trends for ASV richness and bacterial loads were discussed on the basis of
seasonality in the discussion (paragraph starting line 300). However, these differences also
appeared on a diurnal scale which wasn't discussed. Why might these trends be different (e.g.
is bacterial load driven by one dominant ASV that reduced richness?)

We believe so. This pattern is likely in part due to the spike in *Clostridium* pushing
abundance of rare taxa down beyond our detection level. This is supported by the fact that
Shannon diversity doesn't really change across the day, showing that the lower observed
richness in the morning is generated by the drop in rare taxa. We now specifically state this in
the results when reporting on Shannon diversity (L 166). However, a peak in alpha diversity
at noon has also been shown in humans, suggesting at least part of the peak in ASV richness
at noon is due to feeding and not necessarily just a statistical effect.

We have acknowledged this negative relationship in the results (L159), yet unfortunately we
do not have the space to discuss this relationship in detail given we have already expanded
our discussion greatly and have hit the word limit. Temporal dynamics in alpha diversity are
really weak, and also our models of alpha diversity also have quite low explanatory power.
As such, we do not focus that heavily on this aspect of the results. Nevertheless, we do now
compare our results to a study on humans (which didn't account for bacterial load) to show
that this pattern is likely not an artefact:

L314: "Our findings are in line with those from laboratory mice and humans, which also
report spikes in bacterial load when mice become active at dusk^{6,18}, and a peak in alpha
diversity at noon in humans⁷."

- There appeared to be little sampling during the mid-day period (e.g. Figs 2b, 4a). Why was
this and how might this impact results? This should be discussed a bit.

We agree that this situation was not ideal. The reason for the gap is because both meerkats
and humans are not active during the middle of the day, although in winter it is very possible
that meerkats are active but just not monitored (there is now evidence for this from
unpublished accelerometer data). As outlined in point 3 in the summary above, we have
added 80 more samples to try and cover this gap, which does not close it completely but does
narrow the midday gap from 5 hours to 3 hours. We also carry out a sensitivity analysis by
randomly subsampling samples equally across the day, which shows similar results
(presented in Figs S10 and S11). In addition, the smoothing function we use (cubic regression
splines), only fit smoothing knots where there are data (opposed to distributing the knots
equally), so that gaps in the data do not produce spurious trends. We have added this
information in the methods:

L543: "Cubic regression splines calculate smoothing knots based on data density (rather than
distributing them equally along a gradient), and therefore periods of missing data, e.g. during
the middle of the day, do not contain knots nor generate erratic trends."

We also outline our sensitivity analysis for unequal sampling distribution in the methods:

L556: "because samples were collected unequally across the day, with few samples in the
middle of the day, we randomly subsampled 20 samples per hour interval (minus two hours at
noon that had fewer than 20 samples and were therefore excluded) and reran models on the
reduced dataset, and found results were robust to sampling distribution (Fig. S10a)."

Smaller changes

Introduction

- Line 51: There seems to be a typo here. May-September is listed as both summer and winter
207 months

Thanks – fixed (L87)

“The Kalahari region is also highly seasonal, with the climate marked by high temperatures
and sporadic rainfall during the wet summer (October to April), and dry winters (May to
September; Fig. 1c) being cool with almost no rainfall.”

- Line 57: Different storage methods should be added here, as not all samples were frozen
afterwards?

As outlined above in the summary of changes, we have brought the effect of storage up so
that it is the first section of the results (L114). We have added this information to the
introduction as requested (L99):

“For long-term storage, samples prior to 2008 were mostly frozen at -80c (n = 461), or, after
2008, freeze-dried and kept at room temperature (n = 648; Fig. S1a).”

Results

- Line 98: change “genera” to “genus”.

This paragraph has been deleted, and replaced with a methods validation section.

- Line 159: Was this 62% of the 34% of overall variation explained above? Or 62% of overall
variation?

62% of overall variation. However, we have taken out this paragraph since the contribution of
each axis is marked in the figures (Figures 2 and 3).

- Line 183: this header indicates results regarding clostridium, but I don't think there is any
mention of clostridium in this section? Consider rephrasing

Thanks. We have now restructured the results section, and taken out Clostridium in the
headings.

- Line 202: please explain in a bit more detail how genera were identified as core. In most
cases, they are taxa present in a certain number of samples. Is that what is meant by 80%
prevalence here, or does this refer to relative abundance of the taxa? Is the N here the number
of genera or number of samples? A list of the taxa somewhere would be helpful.

Thanks for highlighting this. We agree this was not clear, and in retrospect, we believe we
should have expanded our analyses to include more taxa. We still do focus on 16 genera,

which we call ‘focus genera’ instead of ‘core taxa’. In the previous manuscript, we did limit
our analyses to the most prevalent taxa (> 80% prevalence across samples) because a) they
are the most common and therefore contributing the most to composition (confusingly, these
taxa do make up around 80% relative abundance too, although we did not report this before);
and b) it is much easier to model prevalent taxa to avoid zero inflation. Rare taxa (ie low
prevalence taxa) are very challenging to model.

However, this strategy does risk missing important associations with rarer taxa. For example,
we were particularly interested in identifying juvenile-associated genera, or genera which
increases in the afternoon. Yet we cannot present GAMM models for a hundred or more
genera, this is overwhelming for the reader and not very focused. We therefore compromise:
we present non-parametric differential abundance analyses for all genera with over 15%
prevalence (Fig. S4). This provides a broad summary of which and how many genera are
undergoing temporal changes. However, we still focus on the most common genera for more
in depth analysis (n = 12), and use the differential abundance analysis to select four additional
(but rarer) genera that are showing notable changes to run GAMMs on.

We now explain this in more detail in the manuscript:

L203: “We first performed simple differential abundance non-parametric tests across all
genera with over 15% prevalence across samples (n = 117) to identify genera that were
differentially abundant in the morning compared to afternoon, in the dry season compared to
the wet season, young meerkats versus adults, and adult meerkats versus old meerkats (Fig.
S4). Almost all genera were significantly associated with time of day (Fig. S4a), suggesting
that diurnal oscillations are widespread across gut microbiome members. Only a few genera
significantly differed between dry and wet seasons (Fig. S4b). A small number of genera
were differentially abundant in adults compared to young meerkats (Fig. S4c), whilst none
were differentially abundant in old meerkats compared to adults (Fig. S4d).

We next focused on 16 notable genera in order to model their temporal dynamics using
GAMMs whilst controlling for potentially confounding methodological variables. We
focused on the most prevalent and abundant genera (n = 12) which all had at least 60%
prevalence across samples and together accounted for 75% relative abundance. However, we
used the results from the differential abundance analysis to select four additional rarer genera
that exhibited notable trends for additional analysis, including *Raoultibacter* (43%
prevalence), and *Callulomonas* (38% prevalence). We also include a particularly rare genus,
*Eubacterium* (18% prevalence), which was only present in young individuals.”

- Lines 206-211: a supplemental table may be helpful to show exactly which genera were
influenced by which variables, unless they are all listed in figure 6? Maybe that figure should
be cited here.

All associations are now visualised in Figure 5 (which was previously figure 6). Our updated
figure summarises all effect sizes (and whether they are robust to methodology or not). This
figure is now split into effect sizes of temporal variables (Fig. 5a; ie just the strength of their
temporal dynamics across the three temporal scales) and effect size for mechanistic variables
(ie which underlying mechanisms best explain temporal dynamics; Fig. 5b). Note that we

have altered this figure to also include diversity metrics, and – for clarity - it no longer
visualises the effects of individual methodological variables. We decided to remove
methodological variables because 1) they just don't fit; and 2) due to rather large effect sizes
they tend to distort the x-axis scale, making it very hard to see effect sizes of the biological
variables we are interested in. Instead, we visualise the proportion of variation explained by
methodological variables in Fig. 5c, where we have partitioned model R^2 into temporal,
mechanistic, and methodological variation. This figure now acts as an overall summary of all
models presented in the manuscript.

- Line 227: Were only the genera listed as core used for this analysis?

As outlined in detail above, we now no longer limit the analysis to only core taxa.

Methods

- Line 360: some additional information is needed about this storage experiment. How were
the samples handled after collection? Were the samples split into storage treatments from
single individuals or different individuals were used for each type? etc.

Each sample represent a different individual (1 sample per meerkat). As with samples from
wild meerkats, meerkat's were observed defaecating, the sample collected, and immediately
frozen. Samples were frozen immediately after collection We have clarified this in the
methods:

L660- 671: "Whilst the effects of storage can be accounted for statistically, we wanted to
confirm experimentally that the two storage methods used here do not overly affect bacterial
composition. We experimentally tested the effect of freezing versus freeze-drying on overall
bacterial community composition by collecting fresh faecal samples from nine different
captive meerkats housed at the University of Zurich. Faecal samples were frozen immediately
on collection. A subsample of the sample was then freeze-dried, whilst another subsample
remained frozen at -80°C for one week. DNA was extracted and processed following the
same protocols as described above, with the exception that an internal standard was not added
to samples. To analyse the effect on storage on these samples, samples were normalised by
rarefaction and we performed a marginal PERMANOVA on a Weighted Unifrac distance
matrix, including sample ID and storage as terms."

- Line 363: what statistical methods were used to compare beta and alpha diversity between
these groups?

This is outlined above.

- Line 392: what version of QIIME2 was used?

We used version 2020.2. This has been added in L470.

- Line 397: change “that” to “than”

Thanks, corrected.

- Line 421: what units was weight measured in?

Meerkats are weighed in grams. This has been added, but due to space limitations, this part of
the methods has been moved to the supplementary materials (see Supplementary methods
S1).

- Overall for beta and alpha diversity metrics, was rarefaction used to standardize sequencing
depth?

All analyses were carried out on normalised data (scaling to the internal standard) and
controlling for sequencing depth and other methodological variables in the various models (
for both alpha and beta diversity). Since beta diversity results are based on relative
abundances, rarefying the counts actually makes no difference to results (we tested this). We
now make it clearer that all analyses are conducted on scaled reads (L496, L570, L590).

- Line 518: why was MDS used as opposed to NMDS which appears to be more common for
these analyses?

In my experience, one has difficulty converging NMDS ordinations with any large microbial
dataset due to the huge amount of variation. Convergence is usually possible with small
datasets. In our case, NMDS ordinations did not converge, and we did not want to filter the
dataset too heavily. Even with unconverged NMDS ordinations, the effects looked similar to
those presented, yet we did not want to present unconverged models. We have justified our
use of MDS ordination on L590.

Figures/Tables

- Fig 2a is really difficult to glean information from due to the number of samples and
microbial taxa. It may help to distill the number of taxa displayed to smaller number (e.g. top
5-10 most abundant)

We have revised this barplot to represent every half an hour period (suggested by reviewer 2),
and moved it to figure 3. We still retain 16 genera because this then keeps the colour scheme
consistent across plots.

- Fig2 d-e: what cutoffs were used to delineate groups (e.g. what hours were used for
morning/afternoon and months for wet/dry season). This is explained for age but not these
two.

Thanks for pointing out this omission. We use the ‘noon gap’ in the data as a cutoff
threshold for morning/afternoon (</> 7 hours after sunrise), since this is more biologically

meaningful than 12pm as it represents when meerkats finish their morning bout of foraging.
We have stated these thresholds in more detail in the methods (L608-612), as well as the
legend of Figure 2.

L608: “We therefore ran a differential abundance analysis on all genera with over 15%
prevalence to assess any differences in taxa between morning (< 7 hours after sunrise; n =
743) and afternoon (>7 hours after sunrise; n = 366), dry (May-September; n = 418) and wet
seasons (October-April; n = 691), and young (<1 year; n = 385) and old (> 5 years; n = 97)
meerkats.”

- Fig 2 legend: change GAM to GAMM?

Thanks, fixed.

- Figure 3a: it is unclear to me what joint vs. independent represent. This should be explained
in more detail in the text or legend.

In this version we have excluded this hierarchical partitioning analysis. This is because
another reviewer also queried this analysis, and we realised after careful consideration that it
merely duplicates the models we generated to add arrows to our ordination plot. To add the
arrows to our ordination plot in Figure 3b and c, we apply `vegan::envfit()`, which uses linear
models (with permutations) to model variables onto ordination axes. This is essentially the
same as hierarchical partition analysis, except that the latter also divides variation into
independent variation and shared variation (variation explained by multiple variables). Since
shared variation was low and not an issue, we have decided to keep things simple and report
the stats from the `envfit()` analysis, which match the arrows on the ordination.

- Table S3 caption: change “statics” to “statistics”?

Thanks, fixed.

**Reviewer #2 (Remarks to the Author):**

Risley et al. extensively document microbiome variation in wild meerkats, isolating dynamics
associated with circadian rhythms, seasonality, and host demography. The authors leverage
an extensive sample and metadata set to answer questions previously untackled in
microbiome analyses, and they use appropriate mixed effects models to do so. The paper
provides a unique perspective on gut microbiome dynamics in a wild population and is a
significant contribution to the field.

Thank you for this positive assessment!

I have identified four larger issues that should be addressed before publication however and
also provide a list of smaller concerns.

Larger Issues:

Model choice—The authors do a laudable job of describing their GAMM fitting in the
methods, but their use of hierarchical variance partitioning models is much less well
motivated. The R package used is not specified nor are the validation tests. Moreover, the
authors fail to explain (i) why the hierarchical variance partitioning model approach was used
to analyze the biological predictors of the first four major principal components of the MDS
ordination and (ii) why it wasn't used for analyses of biological predictor role for other
dependent variables. Furthermore, why analyze biological fit on 4 MDS axes rather than just
modeling fit of overall dissimilarity with PERMANOVA?

As mentioned in a response to Reviewer 1, we have now removed the hierarchical
partitioning analysis because it is simply not necessary and does more to confuse the matter
(see comment starting L346 of this document). Our motivation for using the hierarchical
partitioning analysis was that we wanted to show the predictors of each ordination axis
independently, given that each axis represents a different suite of taxa and the first four axes
make up a disproportionate amount of the variation. E.g. axis 1 largely represents Clostridium
on one side and Bacillaceae on the other end, whilst axis 2 largely represents Bacteroides.
Most of the diurnal temporal dynamics appears to be driven these taxa, and we wanted to
emphasize this with the hierarchical partitioning analysis. The use of PERMANOVA alone
suggests that diurnal effects are rather weak (although it should be noted that effect sizes are
large, but R² is weak), when in fact the effect sizes are really rather large when considering
the first two axes of ordination.

Instead of hierarchical partitioning analysis, we now simply report the results of the envfit()
models, which map variables onto an ordination using linear models. We have added more
details on these models in the methods.

L595: “We statistically tested for differences in centroids across axes 1 and 2 (Fig. 3b) and 3
and 4 (Fig. 3c) by using the *vegan::envfit* function, controlling for methodological variables,
and with 999 permutations. The *envfit* function uses linear model permutations to map
variables onto an ordination.”

Abundance patterns— The fact that all but one major genera exhibit the same diurnal shifts
despite being associated with different biological and environmental variables is confusing to
me. To what extent may these just be reflecting the overall trend in load? The strong effect of
sequencing depth for all genera except Geodermatophilus seems to indicate this may be at
play. Are similar shifts observed when analyzing relative abundance? (I'm not advocating
including relative abundance analyses in the published paper, just using them as a tool to
better understand why absolute abundance patterns appear as they do.)

Thanks for bringing this up. Previously we only tested genera with over 80% prevalence
(since these were the most abundant and also the most reliable to model). However, in our
new analysis we have checked all taxa with over 15% prevalence using simple non-
parametric differential abundance analysis (which we visualise in Fig. S4). Whilst most still
underwent the same patterns, we did find a few genera which do the opposite and increase in

the afternoon. Because these are biologically interesting, we have included these in our list of
focus genera. See comment starting L239 in this document for more explanation on the 16
genera we model in our revised analysis.

I hope that our new results, which show that there is a number of genera that do the opposite
pattern and increases in the afternoon, satisfies your concern that this might be an analytical
artefact or just a consequence of bacterial load.

The extreme response exhibited by *Clostridium* may in part reflect the fact that it has much
higher (2-5x) 16S copy number than the other genera analyzed (for which genomes are
available and copy number estimates are published in rrnDB). While the authors note in the
methods that the internal standard technique directly assesses copy number but generally
reflects absolute abundance trends, they don't discuss how large differences in copy number
will impact their interpretation.

Thanks for this resource. In response, we have discussed this as a caveat in the last paragraph
in the discussion (L394-414), where we also bring up other methodological limitations.

“Our study combined extensive longitudinal data and microbiome load quantification to
advance our understanding of temporal dynamics in gut microbiomes. Nevertheless, it faces
some study design and methodological limitations that may affect interpretations. Notably,
the use of internal standards is likely prone to high technical variation, since it is challenging
to accurately standardize sample weight, and subsequent technical variation can be inflated
by PCR bias⁴⁵. Our technical replication analysis confirmed that technical variation was
higher for estimates of bacterial load (10%) than measures of alpha and beta diversity (~2%).
Whilst this variation is non-negligible, sample ID still accounted for 90% of variation and
therefore the identification of true biological associations is possible, especially with large
sample sizes. We also minimise the risk of further PCR bias by controlling for sequencing
depth in all analyses⁴⁵. A perhaps more serious concern is that variation in 16S rRNA gene
copy number biases bacterial load estimates due to differences in the number copies between
bacterial species. To date there is no consensus about how to control for 16S copy number in
amplicon data⁴⁶, and bacterial genomes can contain between one and 21 gene copies^{47,48}. As
such, our estimated abundances are almost certainly over-estimates. *Clostridium* species
predictably have high copy numbers (~10 copies), therefore at least part of the large spike in
*Clostridium*, and reflected in bacterial load, may be an artefact of high copy number.
Nevertheless, we are interested in estimating relative changes in abundance over time within
communities, rather than comparing abundances amongst taxonomically different
communities. Therefore, whilst the rates of change over time are not comparable between
different taxa, the overall direction of change for each taxa is reliable.”

It does seem that the *Clostridium* genus has an average 16S copy number of around 10
(compared to average of 5, I believe), and it seems that this copy number is generally quite
consistent across strains. This definitely could explain the very large peak in the morning.
Whilst the overall trend for *Clostridium* should be reliable, it is true that one cannot compare

the degree of change over time between species, and this is why we don't include analyses
such as ecological networks, which can be very biased by differences in copy number.

Because we merge ASVs by genus, and it conceivable that different ASVs represent different
strains with different copy numbers, we additionally checked that all *Clostridium* ASVs (~7
ASVs) were behaving in the same way, and they all exhibited very similar dynamics (not
presented). This supports our decision to merge ASVs by genus, even though the different
ASVs may have different copy numbers.

Diurnal time series—For totally understandable methodological/behavioral reasons, the
authors were unable to sample evenly across the course of the day. However, they fit their
models across the entire day even though between ~5 and ~10 hours after sunrise their
sampling is very sparse. The confidence intervals are somewhat larger there, but I'm not sure
you should be fitting through at all. Certainly more discussion of this limitation is necessary.

As outlined in the summary above, we have added 80 more samples to the dataset to try and
close this gap a little. This gap is now around 2-3 hours, opposed to 5 hours. Whilst still not
ideal, with analytical precautions and sensitivity analyses we believe GAMMs are still
appropriate, with a clear acknowledgement that there is uncertainty around the middle of day.
Firstly, the cubic regression splines we fit only place smoothing knots where there is enough
data so that small sample sizes don't generate erratic trends. Therefore, in places of low data
(including, for example, in very old meerkats where less data is available), the model makes
no assumptions about what is going on but basically just fits a linear trend between periods of
dense data. Given the constraints on the GAMM smooths (cubic regression splines, plus
correlation error distributions, which both limit how 'wobbly' the line can be), it would take
quite aberrant data during this 2 hour period to change the shape of the trend.

As well as adding extra data, we also run random sampling across the day to check whether
uneven sample size distribution (apart from 2 hours in the middle of the day where there was
less than 40 samples) alters results. It doesn't, and we present this in the supplementary
material (Fig S10 and S11). Therefore, we acknowledge that uncertainty is high during the
middle of day, but provide evidence that this gap does not affect estimates for the rest of the
511 day.

In response, we have added the following lines:

L147: (results) "Bacterial load tended to be highest early in the morning and lowest
approximately 10 hours after sunrise (Fig. 2a), although it should be noted there is
considerably uncertainly regarding estimates for the middle of the day when sampling is
sparse."

L543 (methods): "Cubic regression splines calculate smoothing knots based on data density
(rather than distributing them equally along a gradient), and therefore periods of missing data,
e.g. during the middle of the day, do not contain knots nor generate erratic trends."

L556 (methods): "because samples were collected unequally across the day, with few
samples in the middle of the day, we randomly subsampled 20 samples per hour interval

(minus two hours at noon that had fewer than 20 samples and were therefore excluded) and
reran models on the reduced dataset, and found results were robust to sampling distribution
(Fig. S10a).”

It’s unclear to me as a non-meerkat specialist how frequently an individual defecates so
whether there could potentially be samples in the middle of the day if one had access to
burrows or even at multiple times during the day for one host. It is hard to tell from Figure 1
whether there are individuals who were ever sampled multiple times on the same day even
during the two dense sampling campaigns. If there are, would it be possible to provide
supplemental figures plotting their observed richness or load over such a day? This would be
especially helpful for estimating how representative the overall fit is for individuals given
that there is such variance at any given time of sampling.

Unfortunately we do not sample the same individual on the same day. The closest samples for
individuals are about a month or two apart, since our aim was to sample meerkats rather
evenly throughout their lives. In this system, there appears to be very little ‘individual’ effect,
although samples collected close together (~ within a few months) are more similar than
samples collected further apart (> a year). This individual stability (and predictors of
stability) is actually the subject of our next paper on this dataset, therefore this is in part why
we do not focus much here on individual effects. The effect of ID as a random effect in all
the models we present is almost always not significant (with the exception of alpha diversity,
where there is some small effect of ID). There is an effect of ID on beta dissimilarity ($r^2 =$
20%), but the effect size is very small (ie, individual centroids are very close together).
Therefore, whilst individual effects are probably much larger over the short term, over the
long term (years), individual effects are very weak. This is line with the recent Grieniesen et
al. paper (Science, 2021) on baboon microbiomes over long time frames.

Dominant genera—The specification of “dominant genera” for the enterotype analyses
(Figure 3) needs to be better defined in the results and the methods. Is it just the most
abundant genus? Why aren’t these the genera of focus for the core genera analyses if they are
what distinguishes between communities? At the very least the color schemes should be
consistent between 3b and 4e-h so the reader can more easily tie them together.

The dominant genus is indeed just the most abundant genus in each sample. We now simply
say that points are coloured and grouped “by the most abundant genus in each sample”
(legend Fig. 3). These genera were included in the list of ‘core’ genera in the previous
version. However, to increase clarity the same 16 focus genera are now the same colours
across all figures.

Smaller Concerns:

-line 39 It is unclear what the definition of “biological systems” is in this context. Do you
mean non-microbiome host physiological programs?

Yes. We have clarified this to ‘host physiological circadian rhythms’ (L62).
-line 47-50 What are animals doing when they aren’t foraging?
Resting, playing, moving, etc. They often move very far from their burrows to forage,
therefore they spend quite a lot of time on the move. In the summer, they don’t spend lot of
time foraging (just v early and v late in the day) and this is confirmed by unpublished
accelerometer data.

-line 65 Specify you mean the abundance of each of the 15 genera, not the 15 together.

We have revised this aim to “To identify which genera exhibit predictable dynamics at each
scale” (L108).
-line 68 What kind of models?
This section has now been removed. We now explain model structure at the point where we
present the results of the model in question, rather than try and summarise our models at the
end of the introduction.

-lines 98, 99 Specify if 60% and 30% are means and provide standard deviation or confidence
intervals for the summary statistic.

We have now replaced this section with the results of our investigation on storage methods
and the technical replication analysis, which we believe is more pertinent and important than
a summary of relative abundances.
-The presentation of the GAMM model fits is a bit hard to follow in the text (e.g. lines 106-
116; 144-156). Is there a way to report statistical significance for relevant variables in the text
rather than just directing to a supplementary table?

We now add all relevant statistics within the text, as well as in the supplementary table.
-line 469-470 You’re testing underlying mechanism not necessarily “identifying” them since
you don’t have all possible relevant mechanistic variables included in your analyses.

Noted, we have altered this terminology across the manuscript.

-line 509-512 What would’ve been evidence for more than marginal overfitting? Isn’t a ¼
drop in variance explained a lot?

Good question. We performed a one sample t-test (we only present this in the R markdown
report) to test the R2 from our model and 100 train/test models and it was significantly
different. We have altered this in the text:

L581: “Finally, we validated the model by splitting the dataset into training and test sets 100
600 times. The model explained on average 24% of variation in the untrained data (in comparison
to the 29% reported), indicating model predictions were likely over-fitted and true
explanatory power was closer to ~24%.”

-line 522-528 You could analyze with `adonis2` function in `vegan`, rather than `adonis` function,
to capitalize on a marginal sums of squares approach (`by="margin"`) and thus not be subject
to the same sequential effects.

We have now done this. The PERMANOVA results are the marginal effects. The
PERMANOVA results are outlined in Table S4.

Figures

-Please provide supplementary figures with histograms of your sample set. For example, what
is the frequency of samples for time of day (maybe hourly bins), months, and age? And what
is the frequency of samples for time of day by age groups (underlying the figure 5 analysis)?
It's not possible to back this out of Figure 1 or the highly dense observed data plots (like 2b-
615 d), but such information is useful for understanding how robust the patterns are.

We have added the histograms of each temporal scale in Figure 1e. These were previously
just in the R markdown report. We have also added histograms to the top of Figure 6 (which
was figure 5 – diurnal oscillations by age group).

-I find figure 2a really hard to parse, especially since the x-axis is not plotted as a continuous
axis. Either fewer families need to be plotted and/or averages for ordinal time points need to
be plotted. The current version could be included in the supplement if you think it is
necessary, but it is too busy to read as is. Also, 2A should not be cited after the sentence “The
most abundant genera across samples was *Clostridium sensu stricto* 1, an anaerobe that made
up 30% of reads, and which was more abundant in the morning than the afternoon and
evening” since there are no genera results in it.

We have now altered this figure to represent mean composition per half hour interval (Figure
3a below). We have also removed the paragraph you refer to here (which previously just
summarised overall composition).

New figure 3:

-“Dominant genera” needs to be defined in the legend of Figure 3. Is “other” any other genera
or a group of non-specified but not exhaustive genera?

We have clarified that dominant means the most abundant genus per sample. “Other” means
that sample was dominated by a genus not listed in the colour key. We have clarified these
points in the legend.

-Figure 4d x and y axes should match, the axes should not have decimal places in the
superscript, and the blue line should be defined (is it a fit line or 1:1)?

We have removed this figure in this version, to save space, since we believe it is not essential.

-Why are 4h-j not also on a log scale?

We have now visualised this figure on the log scale (Fig. 4b).

-Can you provide the overall load oscillations for each age group in Figure 5? If differences

in overall abundance are underlying much of the genera oscillations (see “Abundance
 Patterns” above) we’d expect to see a similar overall abundance curve for each age group.
 But if the overall signal varies while the genera patterns are consistent, that supports the idea
 that genera behave somewhat independently.

We have added bacterial load to Figure 4b (see below). We do not believe that bacterial load
 is independent of these taxa dynamics, since by definition bacterial load is the sum off all
 taxa. However, in the revised figure below, you can see that genera are demonstrating
 different dynamics, providing some evidence that these are somewhat independent of
 bacterial load.

New figure 4:

-Figure 6 in general is an excellent way of summarizing some very complex model results. It
 would be helpful though to have a legend on the plot specifying what dot size indicates.

Thank you. We have actually now expanded this figure to include diversity measures,
 visualized effect sizes on the same axis (so that it is easier to compare effect sizes), and
 distinguished between robust and non-robust effects (ie, effects that show the same trends
 across frozen and freeze-dried samples).

**Reviewer #3 (Remarks to the Author):**

The manuscript titled “Diurnal oscillations in gut microbiome load and composition eclipse
 seasonal and lifetime dynamics in wild meerkats, *Suricata suricatta*” aims to quantify and
 compare the meerkat gut microbiome across diurnal, seasonal, and lifetime cycles.

The general approach of the study was to characterize the bacterial load and structure of the
meerkat gut microbiome using 1027 samples collected in the Kalahari across 20 years of
morning and evening behavioral observations of individually known meerkats. Longitudinal
sampling was available for 168 meerkats. The gut microbiome was characterized via 16S
rRNA gene sequencing. Bacterial load was estimated by scaling reads to internal standards
(commercially available) that were introduced prior to DNA extraction. Alpha and beta
analyses were performed.

The key finding of the study was that most common bacterial genera exhibited diurnal
oscillations in bacterial load. These oscillations were driven by changes to *Clostridium sensu*
*stricto* 1, and were better explained by light-dark cycles than foraging schedule. Diurnal
cycling of the microbiome did not decay with age.

The conclusion of the study was that diurnal oscillations shape gut microbiome load and
structure in wild meerkats.

Comments/Concerns

This is a high-quality and well written study addressing circadian rhythms of the gut
microbiome in a wild population across 20 years of intensive sampling and behavioral
observation. The breadth, scope, and detail of the metadata available for the samples is
impressive. Commendably, the authors provide all code in an Rmarkdown file. Extraction
and PCR negative controls were included on all sequencing runs.

Thank you for this positive assessment of our study.

I initially had two concerns, however, both are ultimately addressed in the study.

1) The first potential concern was that fecal samples collected prior to 2008 were frozen at -
80°C, while those collected after 2008 were freeze-dried and stored at room temperature.
However, the study includes an analysis showing that biological variation exceeds technical
variation potentially introduced by these differences in storage method.

2) The second potential concern was that bacterial load was estimated using ZymoBIOMICS
Spike-in Controls. I have never seen this approach for quantifying bacterial load and was
instead expecting quantitative real-time PCR. However, after reading the product literature
and related manuscripts, I appreciate the approach. As noted below, I would however
recommend that the approach be described in greater detail in the Methods, as it is novel.

I therefore have only minor suggestions.

Minor comments/edits:

Additional proof-reading is recommended prior to publication (e.g., plural/singular).

We have hopefully done a more thorough job of proof reading in this version, and had a
couple of colleagues read through it too.

Abstract

N/A

Introduction

It would be valuable to present what is known of the mongoose gut microbiome from prior
studies in the Introduction. At present, there is no indication that mongoose gut microbiome
has been characterized at all. If it has not, make this clear.

Thank you for this suggestion. The meerkat gut microbiome has not been characterised
previously, but this comment did lead to a more thorough review of papers on mongoose
microbiomes which lead to some interesting comparisons which we bring up in the
discussion. Notably, there is one study on the Egyptian mongoose gut microbiome that
looked at differences between juveniles and adults.

We did attempt to add this information (ie overview of the previous literature on mongoose
microbiomes) into the introduction, yet despite this we could not find a place to insert this
information that did not disrupt the flow of the paper. Our manuscript is on temporal
dynamics of the meerkat microbiome rather than the characterization of the mongoose gut
microbiome, and we do not believe that outlining findings from the Egyptian mongoose is
necessarily relevant. We have substantially increased the length of the introduction and
discussion, and added additional analyses, and at this point we are at the maximum word
limit. Any inclusion of previous studies on mongooses would require an explanation and
justification on why this is relevant to our study, which took the word count over the limit.

Nevertheless, we do outline some results from the one available study on the Egyptian
Mongoose in the discussion, since this study did a analysis of age (juvenile/adult) on the gut
microbiome and identified one Genus (*Eubacterium*) which was more abundant in juveniles,
which matched our results. However, this is also the case of humans, suggesting that this
process is not necessarily limited to mongooses.

L362: “We also identify some genera that change over juvenile development, including a
decrease in *Eubacterium* over the first year of life. This genus was also more abundant in
juveniles in the Egyptian mongoose³⁸, and is associated with the transitional state between
the infant and adult gut microbiota in humans^{11,39,40}. Therefore, *Eubacterium* likely represents
the weaning period, when young meerkats transition from a milk-based to an arthropod diet.”

Materials and Methods

Line 379 – Did the kit extract all genomic DNA, including that of the host, or only of the
bacteria?

Presumably some DNA of the host was extracted, but likely not very much. Since it would
not have been amplified, we did not test for nor detect host DNA.

Lines 388-390 – Were the DNA extractions also performed in a randomized manner?

DNA extractions were carried out in the order of sequencing, since samples were randomised
after subsampling. We have added this information in L467.

Line 393 – What parameters were used for the DADA2 pipeline? They were not included in
the Rmarkdown report.

We have added parameter information on L472.

“All sequence reads were processed using QIIME2 version 2020.2⁵⁴. Sequences were
merged, quality filtered, and chimera filtered using the DADA2 pipeline²⁹ to generate
amplicon sequence variants (ASVs)^{29,55}. Primers were trimmed and reads were truncated at
244 (forward) and 235 (reverse) base pairs.”

Line 402 – What functions/parameters were used for Decontam? They were not included in
the Rmarkdown report.

We used the `isContaminant` function using the ‘prevalence’ method. We have added this
information in (L481).

Line 408 – By what manner were these ASVs identified as laboratory contaminants? Given
their rarity, they would not affect any study outcomes, but given the careful methodology
used in the study, stating these criteria would benefit others in conducting similar studies.

We also used the *decontam* package for this, using the negative controls. This has been
clarified in L487.

Line 413 – Please explain how samples were scaled to *Allobacillus*. This is a new technique,
which warrants further explanation.

We have added at L493:

“The sample scaling factor was generated by multiplying the mean read count of *Allobacillus*
by its read count in each sample, and sample reads were then multiplied by the sample
scaling factor to normalise the dataset”.

Lines 424-426 – Explain how weights were collected in a near daily manner, or include a
reference to prior descriptions of the process.

Individual meerkats are weighed daily by enticing them onto electronic scales using crumbs
of hard-boiled egg. We have added this information in the relevant section on how we
calculated body condition, which we have had to move to supplementary materials due to
word limit constraints.

Results

Figure 2 – There are two “d” panels in the figure.

Yes I realised this after submitting! The figures have been modified but they are now
labelled correctly.

Discussion & Conclusions

A section on the strengths and limitations of the study would be valuable.

We have expanded the introduction and the discussion substantially. We have added a
paragraph on the most important limitations at the end of the discussion, which we believe
are the technical variation for bacterial load, and 16S copy number. Whilst unequal sampling
distribution is also an unavailable limitation, we have gone to lengths in the methods to show
that this does not affect overall conclusions.

L394: “Our study combined extensive longitudinal data and microbiome load quantification
to advance our understanding of temporal dynamics in gut microbiomes. Nevertheless, it
faces some study design and methodological limitations that may affect interpretations.
Notably, the use of internal standards is likely prone to high technical variation, since it is
challenging to accurately standardize sample weight, and subsequent technical variation can
be inflated by PCR bias⁴⁵. Our technical replication analysis confirmed that technical
variation was higher for estimates of bacterial load (10%) than measures of alpha and beta
diversity (~2%). Whilst this variation is non-negligible, sample ID still accounted for 90% of
variation and therefore the identification of true biological associations is possible, especially
with large sample sizes. We also minimise the risk of further PCR bias by controlling for
sequencing depth in all analyses⁴⁵. A perhaps more serious concern is that variation in 16S
rRNA gene copy number biases bacterial load estimates due to differences in the number
copies between bacterial species. To date there is no consensus about how to control for 16S
copy number in amplicon data⁴⁶, and bacterial genomes can contain between one and 21
gene copies^{47,48}. As such, our estimated abundances are almost certainly over-estimates.
*Clostridium* species predictably have high copy numbers (~10 copies), therefore at least part
of the large spike in *Clostridium*, and reflected in bacterial load, may be an artefact of high
copy number. Nevertheless, we are interested in estimating relative changes in abundance
over time within communities, rather than comparing abundances amongst taxonomically
different communities. Therefore, whilst the rates of change over time are not comparable
between different taxa, the overall direction of change for each taxa is reliable. “

Other

It appears that Ben Dantzer’s name is misspelled in the Acknowledgements, unless that is not
who is being referred to.

Thanks, corrected.

Kevin R. Theis

Reviewers' Comments:

Reviewer #1:

Remarks to the Author:

I have reviewed the author's response to reviewers and the revised manuscript and feel that all my previous comments and concerns were successfully addressed. Therefore, I do not have anymore suggestions. Thank you to the authors for their thorough revision.

Reviewer #2:

Remarks to the Author:

The authors did an excellent job and the manuscript is much improved following revision. The analyses are more clearly explained and interpreted, and the figures highlight the findings in an easily interpreted manner. The results remain highly interesting and valuable to the field, so I am pleased to have gotten to review them and thus learn about them early.

Three very minor comments that could be addressed in proofing:

-line 88 has a typo at the beginning of the sentence. Do you mean "Meerkat diet diversity" instead of "Meerkats diversity"

-The legend for Figure 1C should clarify what time period the average climate data comes from. The years 97-2020 sampled for this study or all years studied at the field station? Presumably the climate has been changing there as elsewhere and while there's no need to show those trends, you just need to be clear where the averages are coming from.

-in lines 57-59 and 361-363 you state human infant gut alpha diversity is higher than other ages. This is not accurate, typically it is considered to be lower than adults (including in citations 11 and 37 referenced here) although beta-diversity is higher. You'll need to clarify what "gut microbiome of infants tend to be more diverse" than if not adults or remove the second clause of that sentence in 57-59 and adjust accordingly in the discussion as well.

Reviewer #3:

Remarks to the Author:

In revising the manuscript the authors have addressed my prior concerns. The new analyses are a welcome addition and the current manuscript is a valuable contribution to the field.

Response to reviewers

We are happy that the three reviewers found our revisions satisfactory. Reviewers 1 and 3 had no further suggestions, and reviewer 2 had some minor suggestions. Below we respond to these. We have also attached our response to the extended comments as a separate document.

Reviewer #2 (Remarks to the Author):

The authors did an excellent job and the manuscript is much improved following revision. The analyses are more clearly explained and interpreted, and the figures highlight the findings in an easily interpreted manner. The results remain highly interesting and valuable to the field, so I am pleased to have gotten to review them and thus learn about them early.

Three very minor comments that could be addressed in proofing:

-line 88 has a typo at the beginning of the sentence. Do you mean "Meerkat diet diversity" instead of "Meerkats diversity"

We have fixed this typo (L89).

-The legend for Figure 1C should clarify what time period the average climate data comes from. The years 97-2020 sampled for this study or all years studied at the field station? Presumably the climate has been changing there as elsewhere and while there's no need to show those trends, you just need to be clear where the averages are coming from.

We have added this information to Fig. 1 legend:

“Seasonal climate across the year measured at the Kalahari Research Station, South Africa, averaged from data between 2009 and 2019”

-in lines 57-59 and 361-363 you state human infant gut alpha diversity is higher than other ages. This is not accurate, typically it is considered to be lower than adults (including in citations 11 and 37 referenced here) although beta-diversity is higher. You'll need to clarify what "gut microbiome of infants tend to be more diverse" than if not adults or remove the second clause of that sentence in 57-59 and adjust accordingly in the discussion as well.

We have changed the statement in the introduction to (L58):

“In humans, microbiome alpha diversity increases over infancy¹¹, whereas it decreases in chimpanzees²⁰, although the gut microbiome of infants tends to have higher inter-individual variation in both species”.

In the discussion, we have clarified that we are referring to alpha diversity (L369):

“Nevertheless, we do report higher variation in alpha diversity in younger meerkats than older meerkats.”